# TopGQ: Post-Training Quantization for GNNs via Topology Based Node Grouping

## Abstract

Graph neural networks (GNN) suffer from large computational and memory costs in processing large graph data on resource-constrained devices. One effective solution to reduce costs is neural network quantization, replacing complex high-bit operations with efficient low-bit operations. However, to recover from the error induced by lower precision, existing methods require extensive computational costs for retraining. In this circumstance, we propose TopGQ, the first post-training quantization (PTQ) for GNNs, enabling an order of magnitude faster quantization without backpropagation. We analyze the feature magnitude of vertices and observe that it is correlated to the topology regarding their neighboring vertices. From these findings, TopGQ proposes to group vertices with similar topology information of inward degree and localized Wiener index to share quantization parameters within the group. Then, TopGQ absorbs the group-wise scale into the adjacency matrix for efficient inference by enabling quantized matrix multiplication of node-wise quantized features. The results show that TopGQ outperforms SOTA GNN quantization methods in performance with a significantly faster quantization speed.

## 1 Introduction

Graph neural networks (GNNs) attract a great amount of attention due to their ability to process diverse unstructured data. They have achieved success in many areas such as recommendation systems (Pal et al., 2020; Fan et al., 2019; Zhang et al., 2023), molecular interaction (Wale et al., 2008; Borgwardt et al., 2005), transportation networks (Bai et al., 2020; Cao et al., 2020), and social network analysis (Qiu et al., 2018; Arazzi et al., 2023). However, GNNs often suffer from substantial computational and memory costs due to increasing demands for processing large graphs, especially on resource-constrained devices.

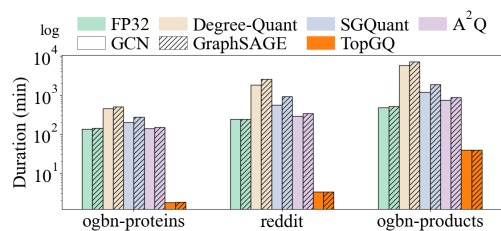

Figure 1: Comparing duration of existing GNN quantization methods against TopGQ.

One promising direction to circumvent this issue is neural network quantization (Choukroun et al., 2019; Zhao et al., 2019; Choi et al., 2021), which reduces the computation and memory requirements of GNN inference by utilizing reduced numerical precision for computations. However, despite its advantages, quantizing GNNs is considered difficult due to the extremely diverse vertex feature magnitudes caused by the message-passing of the GNN algorithm. Since the quantization process is known to be highly sensitive to the magnitude outliers (Wei et al., 2022), such diversity in aggregated features in GNNs results in high quantization errors.

To handle the magnitude outlier problem, several methods (Tailor et al., 2020; Zhu et al., 2022) have been proposed to adopt quantization-aware training (QAT). However, QAT methods accompany significant computation and memory costs for the quantization process, requiring excessive resources and time larger than full-precision pretraining target GNN architecture. Figure 1 reports the time it takes to quantize common GNN architectures using prior GNN quantization methods (Tailor et al., 2020; Zhu et al., 2022). Measured in wall-clock time, the quantization time easily exceeds 100 minutes, and even up to 4.9 days (ogbn-products, Degree-Quant) when the graph size increases.

Real-world graphs, such as traffic networks (Jiang & Luo, 2022; Sharma et al., 2023) or social media graphs (Gao et al., 2023; Yao et al., 2023), are continuously updated, which requires multiple compressing of the GNNs to keep the application up-to-date. Also, in safety-critical applications such as edge fraud detection in financial transactions (Lu et al., 2022; Zhou et al., 2021; Liu et al., 2018), a rapid update is required to avoid safety crisis, and fast quantization of GNN can be necessary.

To this end, we propose TopGQ, the first accurate post-training quantization (PTQ) framework for GNNs addressing the aforementioned issues of existing GNN quantization methods. TopGQ does not involve any form of gradient computation or parameter updates, which makes the proposed method significantly faster compared to the baseline and requires low memory consumption. Instead, TopGQ focuses on the local topological information of the graphs to determine accurate quantization parameters. As a result, our method TopGQ shortens the quantization time by an order of magnitude, making GNN quantizations much faster and more efficient.

The key to achieving high performance is to well utilize the local topology of the graphs. We observe that the existing method relying on the indegree of the vertices is insufficient to capture the diversity of the feature magnitudes. Instead, we propose topology-based node grouping. Because the magnitude of a node feature is determined by local neighbors, we arrange the vertices into several quantization groups that share similar indegree and local Wiener index. In the process, we devise an efficient algorithm for computing the local Wiener index. Lastly, we additionally provide scale absorption method to enable efficient integer matrix multiplication of node-wise quantized feature matrix. The experimental results show that TopGQ outperforms the existing SOTA method for GNN quantization with up to $358\times$ speedups with better or comparable accuracy.

Our contributions can be summarized as follows:

- We show that the magnitudes of node features in GNN are correlated with local topological information from degree centrality and Wiener index.
- We propose a topology-based node grouping, which groups vertices with similar topological characteristics to reduce quantization error from high feature magnitude variance of GNN.
- We propose scale absorption to enable efficient integer arithmetic of node-wise quantized GNN operation by absorbing node-wise scale into an adjacency matrix.
- We propose the first PTQ method for GNN, which outperforms the existing training-based quantization method with meaningful margins while bringing up to $358\times$ less quantization time compared to the baselines.

## 2 PRELIMINARIES

### 2.1 GRAPH NEURAL NETWORKS

Let graph $G = (V, E)$, where $V$ is a set of vertices and $E$ is a set of edges. Each vertex $v_i$ consists of feature vector $h_i$ and adjacency matrix is $A \in \mathbb{R}^{n \times n}$ for $n$ vertices, where $A_{i,j} = e_{i,j}$, if $e_{i,j} \in E$, else 0. To embed topological information in vertex feature, GNN gathers information from neighboring vertices $u_j \in \mathcal{N}(v_i)$ to update hidden vertex feature $h_i$ of $v_i$, which is called the message-passing algorithm. The message-passing algorithm consists of two parts: combination and aggregation. Firstly, hidden vertex feature $h_i^{(l)}$ is multiplied with weight matrix $W^{(l)}$ of $l$-th GNN layer (combination), then the hidden vertex feature $h_i^{(l)}$ of $v_i$ is updated (aggregation) as following:

$$h_i^{(l+1)} = \phi(Wh_i^{(l)}, \bigoplus_{j \in \mathcal{N}(i)} e_{i,j} Wh_j^{(l)}), \tag{1}$$

where $\phi$ feature update operator and $\bigoplus$ is a permutation-invariant aggregation, such as sum or mean.

GNN computation can be represented by multiplications of vertex feature matrix $X \in \mathbb{R}^{n \times d_{in}} = [h_1, \cdots, h_n]^T$, weight matrix $W \in \mathbb{R}^{d_{in} \times d_{out}}$, and adjacency matrix $\tilde{A} \in \mathbb{R}^{n \times n}$ as follows:

$$X_{comb}^{(l)} = W \cdot X^{(l)}, \tag{2}$$

$$X^{(l+1)} = \sigma(\tilde{A} \cdot X_{comb}^{(l)}), \tag{3}$$

where $\sigma$ is nonlinear operation, $\tilde{A}$ may vary with GNN architecture, e.g., GCN (Kipf & Welling, 2016) utilizes normalized graph laplacian matrix $\tilde{A} = D^{-1/2}AD^{-1/2}$, while GIN (Xu et al., 2019) uses binary adjacency matrix $\tilde{A} = A$. GraphSAGE (Hamilton et al., 2017) differs in the aggregation phase by sampling a subset of neighboring vertices instead of considering all neighbors.

## 2.2 QUANTIZATION

Quantization replaces high-bit floating-point operations with low-bit integer operations. We use simple yet effective uniform integer quantization as with scale ($s$) and zero-point ($z$) as follows:

$$x^q = Q(x; s, z) = clamp(\lfloor s \cdot x - z \rceil], q_{max}, q_{min}), \tag{4}$$

$$s = (2^k - 1)/(x_{max} - x_{min}), \tag{5}$$

where k is quantization bit, $q_{max}, q_{min}$ is maximum and minimum value of $k$-bit integer representation, and $\lfloor \cdot \rceil$ is rounding operator. Symmetric quantization has the representation range centered with zero ($z = 0$), while asymmetric quantization uses $z = s \cdot x_{min} + 2^{k-1}$. Also, each row or column may have different quantization parameters ($s, z$), which are calculated independently according to quantization dimension, called row-wise and column-wise quantization, respectively.

There are two mainstream types of quantization: Post-training quantization (PTQ) and quantization-aware training (QAT). On the one hand, PTQ goes through a calibration process which adjusts the scale, zero-point, and rounding directions using only a small set of data. Conversely, QAT methods directly apply gradient-based training to explicitly reduce the quantized network's target loss. The major discrepancy between them is that QAT generally incorporates updating weight parameters, while PTQ methods focus on quantization parameters without weight updates and is much faster.

## 2.3 QUANTIZATION OF GRAPH NEURAL NETWORKS

To achieve efficient inference in terms of computational cost and memory requirements, we should consider both the combination phase (Equation (2)) and the aggregation phase (Equation (3)):

$$X_{comb}^{(l)} = Q(W; s_W, z_W) \cdot Q(X^{(l)}; s_{X^{(l)}}, z_{X^{(l)}}), \tag{6}$$

$$X^{(l+1)} = \sigma(Q(\tilde{A}; s_{\tilde{A}}, z_{\tilde{A}}) \cdot Q(X_{comb}^{(l)}; s_{X_{comb}^{(l)}}, z_{X_{comb}^{(l)}})). \tag{7}$$

As we can see in Equation (6) and Equation (7), we have to choose quantization policy for each $W$, $X^{(l)}$, $X_{comb}^{(l)}$, and $\tilde{A}$. These design choices highly affect the final accuracy and inference efficiency of a quantized network, and many baselines choose different policies. Degree-Quant (Tailor et al., 2020) applies per-tensor quantization for all matrices. While this requires the smallest storage, it suffers from outliers as a single outlier element can affect the quantization scale. $A^2Q$ (Zhu et al., 2022) mitigates this issue by applying column-wise ($W$ and $X_{comb}$) and row-wise ($X$ and $\tilde{A}$) quantization. As row- and column-wise quantization can separate quantization parameters of each node and feature dimension, respectively, it is more robust to outliers at the expense of increased cost.

## 3 RELATED WORK

**GNN Quantization** can efficiently reduce extensive computational costs and memory requirements of graph neural networks (Kipf & Welling, 2016; Xu et al., 2019; Veličković et al., 2018), as they suffer from large size of real-world graphs. Degree-Quant (Tailor et al., 2020) is the first work to quantize GNN, using QAT to allow high-degree vertices to retain full-precision features at training and be quantized later for inference. EPQuant (Huang et al., 2022) focuses on reducing high memory costs incurred by processing large graph data with product quantization, along with quantized GNNs. SGQuant (Feng et al., 2020) and $A^2Q$ (Zhu et al., 2022) are also QAT methods targeting GNN architectures, but they differ in that they allow mixed-precision to assign higher bitwidth to high-magnitude vertices. SMP (Wang et al., 2023) tackles the oversmoothing problem when quantizating deep GNNs with a customized message propagation, and (Eliasof et al., 2023) utilizes wavelet transformations in traditional image compression to quantize GNNs. Notably, existing GNN quantization methods adopt QAT, i.e., incorporating gradient-based iterative weight updates, which require significant computational overheads (Figure 1).

**Graph Topology in GNNs** is often integrated during training to help the model effectively learn the structural information (Ji, 2019; Zhang & Lu, 2020; Hu et al., 2022; Wu et al., 2018; You et al., 2021; Brasoveanu et al., 2023). For example, Ji (2019) uses degree centrality to find highly central vertices in their pooling layer as they are more important for effective representation learning. Also, Zhang & Lu (2020) uses betweenness centrality to assign weights to each node during aggregation. Wu et al. (2018); Brasoveanu et al. (2023) uses Wiener index from chemoinformatics as inputs of GNNs to enhance its performance on general tasks. However, these methods do not relate topological information with node feature magnitudes, especially for quantization.

## 4 MOTIVATIONAL STUDY

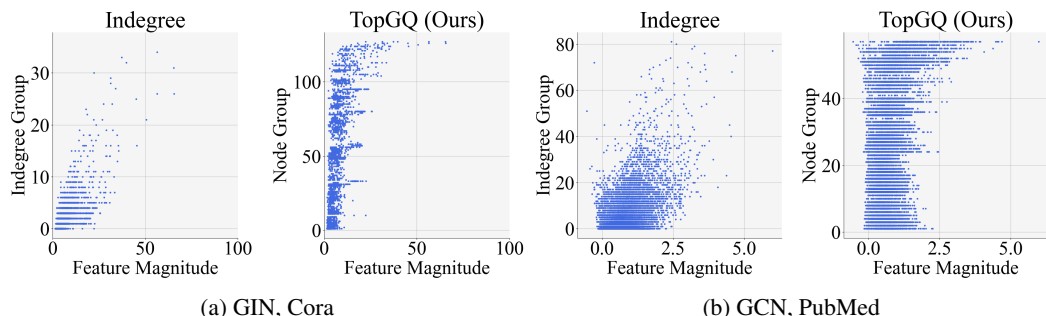

(a) GIN, Cora                    (b) GCN, PubMed

Figure 2: Comparing feature magnitude range of two grouping techniques: indegree (left) and TopGQ (right). For both plots, x-axis denotes feature magnitude and y-axis denotes sorted group index.

For GNNs, the range of the feature values largely depends on each node's structural properties because of their unique message-passing framework. However, existing works (Feng et al., 2020; Tailor et al., 2020) only utilize node indegree which only takes into account 1-hop neighbors. We find that indegree is a suboptimal measure when it comes to determining the quantization group. In Figure 2, we plot the feature magnitude of each quantization group using indegree as the sole metric, and we compare it against the groups used in our method TopGQ. Using only indegree to group the node features, each group tends to have a large range of values with uneven distribution of nodes among the groups. The extreme spread of values within each group would lead to poor representation of the dense region, leading to large quantization errors. Instead, TopGQ proposes to use a topological feature that can better capture such information. Figure 2 shows that each quantization group of TopGQ has a smaller range with a more even distribution across the groups.

## 5 METHODOLOGY

The goal of TopGQ is to rapidly perform quantization with PTQ, while retaining QAT-like performance on GNNs. For this, we propose topology-based node grouping which captures the local topology information into GNN quantization. In this process, we propose a new algorithm to accelerate the computation of the local topology for each node in the graph. Then, we propose scale absorption which allows for efficient integer arithmetic while still preserving the accuracy of node-wise quantization.

### 5.1 QUANTIZATION WITH TOPOLOGY-BASED NODE GROUPING

Figure 3a shows the group generation strategy of TopGQ. Due to the nature of the aggregation phase (Equation (3)), it is evident that the feature magnitudes of a vertex depend greatly on how many and which vertices the features are being aggregated from. As discussed in Section 4, node indegree is a suboptimal measure because it only accounts for a limited amount of information. Instead, we propose to examine the topology of the local subgraph surrounding each vertex, and group the vertices with the same or similar local topology. Within the same group, we can expect that the vertices aggregate similar values and thus sharing a quantization scale leads to minimal quantized errors.

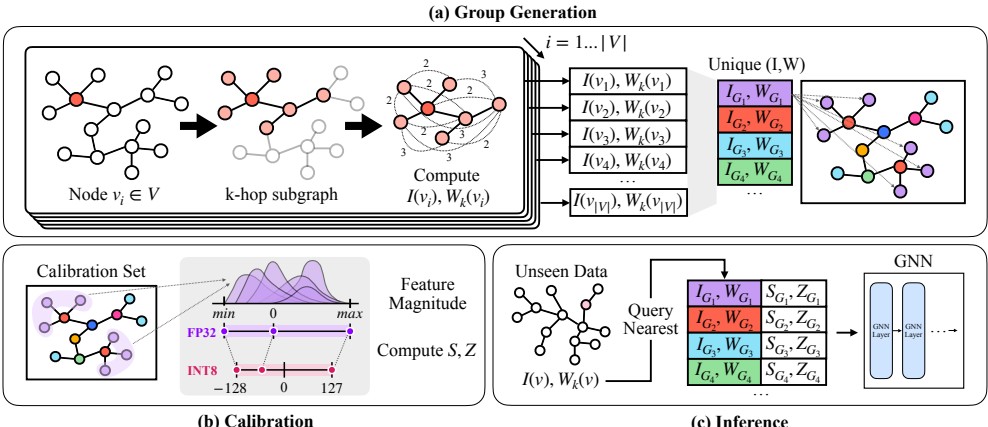

Figure 3: The process of topology-based node grouping. (a) shows group generation using topological characteristics: indegree and Wiener Index. Each color is used to denote each group. (b) shows the calibration process to achieve a set of quantization parameters for each group. (c) demonstrates how inference is done on unseen data by using the quantization parameters of the nearest group.

While there exist several different measures to interpret the topology of a graph, we propose to use Wiener index Graovac & Pisanski (1991) in conjunction with the indegree to better quantify the local structure around each node. Let graph $G = (V, E)$, where $V$ is a set of vertices and $E$ is a set of edges. Wiener index is defined as the sum of shortest lengths between all pairs of vertices:

$$W(G) = \Sigma_{u,v \in V} dis(u, v), \tag{8}$$

where $dis(u, v)$ denotes the shortest path distance between vertices $u$ and $v$. Because Wiener index is originally a graph-level representation, we make an adaptation to use it as a node-level representation. For each vertex, we extract a $k$-hop subgraph around each vertex and compute the Wiener index of the subgraph. Formally put, we define the localized Wiener index $W_k(u)$ of vertex $u$ as:

$$W_k(u) = \Sigma_{v,w \in N_k(u)} dis(v, w), \quad N_k(u) = \{v \in V | dis(u, v) \leq k\}, \tag{9}$$

where $k$ is the predefined hop count and $N_k$ is the set of reachable neighbor vertices within $k$ hops.

Once the localized Wiener index values are obtained for the vertices, we consider indegree $I(u)$ together, and vertices with equal $(I(u), W_k(u))$ are assigned to the same quantization group. For example, in Figure 3a, all purple-colored vertices belong to a single group.

After the groups have been generated, the calibration (Figure 3b) takes place. For each group, group-wise quantization parameters $(S_G, Z_G)$ are obtained according to Section 2.2 by measuring min, max values per group. At inference time (Figure 3c), the test set vertices are assigned to the groups according to their $(I, W)$ pair values. When unseen values are found at inference time, they are assigned to the most similar quantization group by first comparing the $I$ and then the $W$ values.

### 5.2 ACCELERATED COMPUTATION OF LOCALIZED WIENER INDEX

Because the localized Wiener index in Equation (9) requires all-pair shortest paths within the subgraphs, its computation can add a considerable overhead. Although several algo-

---

**Algorithm 1** Accelerated Wiener Index Computation

1: **Input:** Local $k$-hop subgraph $G_{sub} = (V_{sub}, E_{sub})$
2: **Output:** Wiener index $W_k(u) \in \mathbb{N}$
3:
4: **function** addNeighbors(node $u$, set $H$, depth $m$)
5:     **for** $v \in u.nbr()$ **do**
6:         $h_m \leftarrow h_m \cup (u, v)$
7:         **if** $m > 0$ **do**
8:             addNeighbors($v$, $H$, $m - 1$)
9:         **end if**
10:    **end for**
11: **end function**
12:
13: $W \leftarrow k|V_{sub}|^2$
14: $H \leftarrow \{h_l = \varnothing \mid l = 0, ..., k\}$
15: $h_k \leftarrow V_{sub}$
16: **parallel for** node $u \in V_{sub}$
17:    addNeighbors($u$, $H$, $k - 1$)
18: **end for**
19: $W \leftarrow W - \sum_{l=0}^{k} |\bigcup_{i=l}^{k} h_i|$

---

rithms are known for all-pair shorted paths (Dijkstra, 1959; Floyd, 1962; Warshall, 1962; Bellman, 1958; Ford Jr, 1956), they often require substantial computational and space complexity.

For this, we propose a new algorithm to compute the localized Wiener index, shown in Algorithm 1. The key idea is that because we sampled $k$-hop neighbors of a single vertex to extract a subgraph, its diameter (i.e., the maximum distance between two arbitrary nodes) is capped at $2k$. This can be used to efficiently calculate the localized Wiener indices. For instance, $W_2(u)$ is calculated as

$$W_2(u) = |E_{sub}| + 2|d_2| + 3|d_3| + 4|d_4|, \tag{10}$$

where $d_n$ is a set of distance-$n$ node pairs in a local $k$-hop subgraph $G_{sub} = (V_{sub}, E_{sub})$ of node $u$. In practice, obtaining the sets $d_i$ can be time-consuming because this requires costly all-pair shortest paths. Instead, Equation (10) can be restructured in a subtractive manner, using k-hop reachable set $N_i(u)$ that can be easily obtained by simple traversal:

$$W_2(u) = 4 \cdot \Sigma_{v \in V_{sub}} |N_4(v)| - (\Sigma_{i=0}^3 \Sigma_{v \in V_{sub}} |N_i(v)|), \tag{11}$$

From the maximum-value case where all vertices are connected in 4 hops, we subtract the number of occurrences in each of $i$-hop reachable set from the vertices. Additionally, we can substitute some terms trivially obtainable from graph formats such as CSR. The $\Sigma_{v \in V_{sub}} |N_4(v)|$ is simply $|V_{sub}|^2$, $\Sigma_{v \in V_{sub}} |N_1(v)|$ is the number of edges $|E_{sub}|$ and $\Sigma_{v \in V_{sub}} |N_0(v)|$ the number of vertices $|V_{sub}|$:

$$W_2(u) = 4|V_{sub}|^2 - (\Sigma_{v \in V_{sub}} |N_3(v)| + \Sigma_{v \in V_{sub}} |N_2(v)| + |E_{sub}| + |V_{sub}|), \tag{12}$$

The overall process is shown in Algorithm 1. First, we define a function *addNeighbors* (lines 4-11), which recursively adds $l$-hop reachable node pairs into the set $h_l$. Then, we initialize Wiener index $W$ with $k|V_{sub}|^2$ (line 13), and $h_l$ with $\varnothing$. As the computation of *addNeighbors* on node $n$ in $G_{sub}$ is independent of each other, we parallelized the computation (line 16). After the computation, the $h_l$ stores a non-overlapping set of $l$-hop reachable node pairs. By using $h_l$, we calculate $\Sigma_{v \in V_{sub}} |N_l(v)|$ by $|\bigcup_{i=l}^k h_i|$ and obtain the Wiener index result (line 19). Please refer to Table 6 for the experiments on acceleration, compared to Bellman-Ford, Floyd-Warshall, and Dijkstra's algorithm.

## 5.3 INFERENCE FLOW WITH SCALE ABSORPTION

Scale Absorption preserves both the benefits of fixed-point operations and activation precision preservation in TopGQ. Repetitive aggregation in GNN layers may amplify certain values, leading to a node-wise outliers in activations. To prevent the outliers from distorting quantization parameters, TopGQ maintains the same quantization method for the vertex feature matrix $X$ for the combination and aggregation phases. Maintaining node-wise quantization for activation leads to retaining precision, as outliers are isolated from other activation values in the quantization process.

Assuming symmetric quantization for simplicity, it can be represented as $X \approx S_X \cdot X^Q$, where $X^Q$ is the quantized features and $S_X$ is a diagonal matrix of

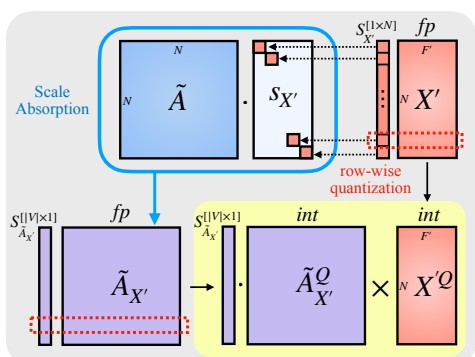

Figure 4: Inference with scale absorption.

the scales of each node group. In the combination phase, the weight parameters are regarded as a single group. Therefore, the quantized form of the combination becomes:

$$X \cdot W \approx S_X \cdot X^Q \cdot W^Q \cdot s_W = (s_W \otimes S_X) \cdot (X^Q \cdot W^Q). \tag{13}$$

For aggregation, using the same quantization method is infeasible, because with quantized $\tilde{A}$ and $X$,

$$\tilde{A} \cdot X \approx S_A \cdot \tilde{A}^Q \cdot S_X \cdot X^Q, \tag{14}$$

which contains the $S_X$ matrix inside the multiplication. Instead, we take advantage of the fact that $\tilde{A}$ is a static topology. After calculating the scale diagonal matrix $S_X$,

$$\tilde{A} \cdot X \approx \tilde{A} \cdot S_X \cdot X^Q = \tilde{A}_X \cdot X^Q \approx S_{A_X} \cdot \tilde{A}_X^Q \cdot X^Q. \tag{15}$$

In the above, the scale diagonal matrix $S_X$ is absorbed into the adjacency matrix $\tilde{A}$ to form $\tilde{A}_X$, which is then row-wisely quantized with the new scale as $S_{A_X} \cdot \tilde{A}_X^Q$. At inference time, this can be pre-calculated as both $\tilde{A}$ and $S_X$ only depend on the topology of the input graph.

Table 1: Performance on node classification task using large graph datasets.

| Dataset | Bit | Method | Type | GCN | | GraphSAGE | |
|---|---|---|---|---|---|---|---|
| | | | | Acc. | Q. Time | Acc. | Q. Time |
| Reddit | FP32 | - | - | 90.60 | - | 94.64 | - |
| | INT4 | Degree-Quant | QAT | 49.25 | (31.18h) | 89.86 | (42.23h) |
| | | SGQuant | QAT | **88.74** | (9.19h) | 63.73 | (15.75h) |
| | | $A^2Q$ | QAT | 58.31 | (4.92h) | 52.65 | (5.78h) |
| | | TopGQ (Ours) | PTQ | 83.95 | **(0.02h)** | **93.93** | **(0.02h)** |
| | INT8 | Degree-Quant | QAT | 90.91 | (30.39h) | 90.35 | (42.49h) |
| | | SGQuant | QAT | 88.67 | (9.46h) | 69.12 | (15.48h) |
| | | $A^2Q$ | QAT | 61.15 | (4.91h) | 76.26 | (5.70h) |
| | | TopGQ (Ours) | PTQ | **91.13** | **(0.02h)** | **94.60** | **(0.02h)** |
| ogbn-proteins | FP32 | - | - | 56.94 | - | 73.33 | - |
| | INT4 | Degree-Quant | QAT | 57.37 | (7.68h) | 50.02 | (8.84h) |
| | | SGQuant | QAT | 52.97 | (3.46h) | 57.77 | (4.64h) |
| | | $A^2Q$ | QAT | 44.95 | (2.35h) | **71.98** | (2.49h) |
| | | TopGQ (Ours) | PTQ | **60.08** | **(0.01h)** | 68.93 | **(0.01h)** |
| | INT8 | Degree-Quant | QAT | **59.32** | (7.49h) | **73.81** | (8.38h) |
| | | SGQuant | QAT | 52.77 | (3.32h) | 69.30 | (4.58h) |
| | | $A^2Q$ | QAT | 44.41 | (2.35h) | 69.38 | (2.51h) |
| | | TopGQ (Ours) | PTQ | 58.05 | **(0.01h)** | 73.34 | **(0.01h)** |
| ogbn-products | FP32 | - | - | 78.41 | - | 71.65 | - |
| | INT4 | Degree-Quant | QAT | **70.58** | (98.38h) | 65.05 | (121.78h) |
| | | SGQuant | QAT | 26.90 | (20.03h) | 27.38 | (37.17h) |
| | | $A^2Q$ | QAT | 23.62 | (13.16h) | 22.21 | (14.69h) |
| | | TopGQ (Ours) | PTQ | 57.55 | **(0.34h)** | **71.02** | **(0.34h)** |
| | INT8 | Degree-Quant | QAT | 75.26 | (95.95h) | 69.18 | (118.96h) |
| | | SGQuant | QAT | 65.71 | (20.18h) | 41.71 | (31.25h) |
| | | $A^2Q$ | QAT | 47.91 | (12.57h) | 58.26 | (14.66h) |
| | | TopGQ (Ours) | PTQ | **76.94** | **(0.34h)** | **73.67** | **(0.34h)** |

# 6 EXPERIMENTS

## 6.1 EXPERIMENTAL SETTINGS

We report evaluation results on two representative tasks: Node-level and graph-level classification. We use Reddit, ogbn-proteins, and ogbn-products, Cora, CiteSeer, and PubMed for node classification task, and use PROTEINS and NCI1 for graph classification task. For baselines, we use three GNN quantization methods: Degree-Quant (Tailor et al., 2020), SGQuant (Feng et al., 2020), and $A^2Q$ (Zhu et al., 2022). To ensure a fair comparison, we use fixed-precision quantization for both SGQuant and $A^2Q$ when attaining experiment results. We use GCN (Kipf & Welling, 2016), GIN (Xu et al., 2019), and GraphSAGE (Hamilton et al., 2017) architectures with 4-bit and 8-bit integer quantization. For a fair comparison, we apply the same bitwidth for all layers. We use $k = 3$ for ogbn-products, PROTEINS, and NCI1 and $k = 2$ for other datasets. More details can be found in the Appendix.

## 6.2 NODE CLASSIFICATION RESULTS

The experimental results of quantization accuracy comparison of node classification task are shown in two settings: larger graphs (Table 1) and more conventional sized graphs (Table 2). The results show that TopGQ performs comparable or significantly better in accuracies, and achieves an order of magnitude faster quantization time. Taking Reddit with 4-bit GraphSAGE as an example, the best-performing baseline is Degree-Quant, with 89.86% accuracy. However, it suffers from almost 42.27 hours of quantization time. SGQuant and $A^2Q$ are faster on quantization, but suffer from severe accuracy drops. On the other hand, TopGQ achieves a significantly higher accuracy of 93.93%, with only 0.02 hours of quantization time. This is more than 1000× faster than Degree-Quant, and more than 100× faster than the low-performing baselines (SGQuant and $A^2Q$).

Table 2 shows results on the smaller graphs that are more commonly used in existing GNN quantization literature. The results show a similar trend overall. TopGQ shows comparable performance compared to the existing baselines with significantly low overhead for GNN quantization. The

Table 2: Performance on node classification task using smaller graph datasets.

| Dataset | Bit | Method | Type | GCN | | GIN | | GraphSAGE | |
|---|---|---|---|---|---|---|---|---|---|
| | | | | Acc. | Q. Time | Acc. | Q. Time | Acc. | Q. Time |
| | FP32 | - | - | 82.08 | - | 78.54 | - | 79.58 | - |
| | INT4 | Degree-Quant | QAT | 79.02 | (9.64s) | 71.90 | (31.47s) | 73.50 | (15.54s) |
| | | SGQuant | QAT | 79.02 | (3.20s) | 70.21 | (4.22s) | 75.30 | (8.62s) |
| | | $A^2Q$ | QAT | 52.68 | (2.09s) | 64.64 | (1.72s) | 74.16 | **(2.53s)** |
| Cora | | TopGQ (Ours) | PTQ | **81.50** | **(1.40s)** | **78.58** | **(0.99s)** | **79.64** | **(0.87s)** |
| | INT8 | Degree-Quant | QAT | 81.80 | (9.82s) | 74.60 | (31.45s) | 77.50 | (15.52s) |
| | | SGQuant | QAT | 80.51 | (3.60s) | 73.32 | (4.53s) | 75.32 | (8.38s) |
| | | $A^2Q$ | QAT | 79.96 | (1.60s) | **78.74** | (1.95s) | 76.12 | (2.48s) |
| | | TopGQ (Ours) | PTQ | **82.08** | **(1.12s)** | 78.42 | **(1.18s)** | **80.30** | **(0.87s)** |
| | FP32 | - | - | 72.34 | - | 70.24 | - | 71.96 | - |
| | INT4 | Degree-Quant | QAT | 22.34 | (21.72s) | 47.92 | (90.57s) | 17.14 | (40.67s) |
| | | SGQuant | QAT | 68.08 | (5.57s) | 46.70 | (8.23s) | 48.34 | (17.91s) |
| | | $A^2Q$ | QAT | 54.00 | (2.08s) | 46.04 | (2.67s) | 66.22 | (3.18s) |
| Citeseer | | TopGQ (Ours) | PTQ | **71.90** | **(1.17s)** | **70.14** | **(1.14s)** | **71.76** | **(1.05s)** |
| | INT8 | Degree-Quant | QAT | 69.72 | (22.03s) | 58.34 | (92.75s) | 69.10 | (40.63s) |
| | | SGQuant | QAT | 68.34 | (5.85s) | 51.30 | (8.56s) | 54.12 | (18.47s) |
| | | $A^2Q$ | QAT | 70.48 | (1.77s) | 67.26 | (2.36s) | 66.04 | (3.15s) |
| | | TopGQ (Ours) | PTQ | **72.28** | **(1.11s)** | **70.26** | **(1.16s)** | **71.96** | **(1.05s)** |
| | FP32 | - | - | 80.32 | - | 78.82 | - | 78.84 | - |
| | INT4 | Degree-Quant | QAT | 78.62 | (21.33s) | 76.56 | (108.07s) | 78.18 | (34.38s) |
| | | SGQuant | QAT | 76.08 | (5.41s) | 65.28 | (8.24s) | 71.08 | (15.86s) |
| | | $A^2Q$ | QAT | 69.72 | (2.17s) | 51.90 | (2.60s) | 73.92 | (3.31s) |
| Pubmed | | TopGQ (Ours) | PTQ | **79.58** | **(1.21s)** | **77.70** | **(1.18s)** | **79.00** | **(1.12s)** |
| | INT8 | Degree-Quant | QAT | 79.24 | (21.56s) | **79.70** | (109.59s) | 78.42 | (34.07s) |
| | | SGQuant | QAT | 78.06 | (5.31s) | 75.22 | (8.91s) | 73.44 | (15.66s) |
| | | $A^2Q$ | QAT | 76.44 | (1.70s) | 76.40 | (2.15s) | 75.36 | (3.24s) |
| | | TopGQ (Ours) | PTQ | **80.30** | **(1.08s)** | 78.62 | **(1.16s)** | **78.94** | **(1.22s)** |

Table 3: Performance on graph classification task.

| Dataset | Bit | Method | Type | GCN | | GIN | | GraphSAGE | |
|---|---|---|---|---|---|---|---|---|---|
| | | | | Acc. | Q. Time | Acc. | Q. Time | Acc. | Q. Time |
| | FP32 | - | - | 76.19 | - | 74.79 | - | 72.87 | - |
| | INT4 | Degree-Quant | QAT | **75.21** | (2158.47s) | 70.44 | (1407.09s) | 63.72 | (1371.54s) |
| | | SGQuant | QAT | 59.84 | (203.70s) | 59.48 | (190.28s) | 59.66 | (249.86s) |
| | | $A^2Q$ | QAT | 71.16 | (128.52s) | 65.59 | (116.96s) | **73.59** | (209.23s) |
| PROTEINS | | TopGQ (Ours) | PTQ | 70.15 | **(4.20s)** | 70.61 | **(3.94s)** | 69.67 | **(4.21s)** |
| | INT8 | Degree-Quant | QAT | 74.93 | (2140.48s) | 69.72 | (1368.98s) | 63.61 | (1358.99s) |
| | | SGQuant | QAT | 72.40 | (203.61s) | 69.73 | (190.71s) | 61.99 | (261.81s) |
| | | $A^2Q$ | QAT | 73.05 | (136.03s) | 66.85 | (129.83s) | 70.62 | (194.75s) |
| | | TopGQ (Ours) | PTQ | **75.94** | **(4.11s)** | **74.86** | **(3.86s)** | **74.00** | **(4.17s)** |
| | FP32 | - | - | 80.41 | - | 81.46 | - | 78.46 | - |
| | INT4 | Degree-Quant | QAT | **73.55** | (4588.48s) | 76.42 | (3110.10s) | 69.46 | (3585.30s) |
| | | SGQuant | QAT | 63.92 | (530.27s) | 53.09 | (571.44s) | 66.13 | (778.96s) |
| | | $A^2Q$ | QAT | 68.81 | (668.52s) | **79.08** | (648.44s) | 72.38 | (656.70s) |
| NCI1 | | TopGQ (Ours) | PTQ | 65.09 | **(9.36s)** | 78.49 | **(8.98s)** | 76.43 | **(9.18s)** |
| | INT8 | Degree-Quant | QAT | 75.47 | (4493.82s) | 77.59 | (3025.24s) | 69.12 | (3449.79s) |
| | | SGQuant | QAT | 68.47 | (527.19s) | 74.36 | (572.13s) | 67.59 | (799.31s) |
| | | $A^2Q$ | QAT | 75.64 | (648.18s) | 79.17 | (635.09s) | 76.86 | (645.50s) |
| | | TopGQ (Ours) | PTQ | **80.91** | **(9.35s)** | **81.88** | **(8.97s)** | **79.16** | **(9.22s)** |

quantization times are relatively short for all methods, which comes from a small number of vertices and edges for the datasets. Nonetheless, TopGQ is the fastest in quantization time in all cases.

Interestingly, TopGQ sometimes outperforms QAT baselines or even the FP32 network. This hints that the existing QAT baselines do not consider the nature of GNN. On the other hand, our method directly integrates the nature of GNN aggregation into the quantization parameters by grouping nodes by their k-hop topological structure. We provide a more detailed analysis in Appendix I.

Table 4: Ablation study of TopGQ.

| Bit | Node Grouping | Scale Absorption | PROTEINS | | | NCI1 | | |
|-----|---------------|------------------|----------|-----|---------------|------|-----|---------------|
| | | | GCN | GIN | Graph SAGE | GCN | GIN | Graph SAGE |
| INT4 | ✗ | ✗ | 57.32 | 45.51 | 44.05 | 53.35 | 60.66 | 73.80 |
| | Indegree | ✗ | 56.15 | 45.04 | 50.65 | 60.54 | 69.71 | 75.46 |
| | L. Wiener Index | ✗ | 61.28 | 47.12 | 62.76 | 60.93 | 72.76 | 75.63 |
| | L. Wiener Index | ✓ | 69.94 | 70.92 | 68.93 | 65.88 | 75.37 | 75.98 |
| INT8 | ✗ | ✗ | 56.14 | 55.91 | 61.25 | 79.63 | 81.29 | 78.30 |
| | Indegree | ✗ | 72.57 | 71.86 | 70.48 | 78.91 | 81.28 | 78.32 |
| | L. Wiener Index | ✗ | 75.64 | 73.94 | 73.69 | 80.89 | 81.90 | 79.18 |
| | L. Wiener Index | ✓ | 75.65 | 74.34 | 72.20 | 79.72 | 81.36 | 78.43 |

Table 6: Comparison of node-wise Wiener index computation time.

| Algorithm | Cora | CiteSeer | Pubmed | PROTEINS | NCI1 | Reddit | ogbn-proteins | ogbn-products |
|-----------|------|----------|--------|----------|------|--------|---------------|---------------|
| Bellman-Ford | 0.35s | 0.46s | 19.87s | 40.07s | 512.13s | 4.21h | 2.84h | 305.50h |
| Floyd-Warshall | 0.15s | 0.21s | 4.68s | 8.50s | 11.27s | 0.57h | 0.41h | 35.44h |
| Dijkstra | 0.19s | 0.29s | 2.70s | 12.75s | 12.49s | 0.16h | 0.11h | 8.52h |
| Ours (§5.2) | **0.02s** | **0.01s** | **0.06s** | **4.84s** | **1.78s** | **0.0004h** | **0.0002h** | **0.2855h** |
| Speed Up | 9.77× | 31.65× | 43.50× | 1.76× | 6.32× | 412.23× | 602.30× | 29.83× |

## 6.3 GRAPH CLASSIFICATION RESULTS

The experimental results on graph classification are in Table 3. The proposed method, TopGQ, significantly improves quantization speed while maintaining competitive classification performance. For instance, Degree-Quant takes almost an hour to quantize the GraphSAGE model on NCI1, with a significant drop in accuracy of 9.0%p. In contrast, TopGQ achieves remarkable speed improvements with PTQ, requiring only about a minute for quantization across all datasets and models. This efficiency highlights the superiority of TopGQ, as it achieves a balance between accuracy and quantization speed, making it a practical choice for large-scale graph-level classification tasks.

## 6.4 ABLATION STUDY

We conducted an ablation study to show the effect of the proposed topological quantization groups, shown in Table 4. The PTQ baseline without any proposed methods suffers from accuracy degradation due to high-variance node-wise magnitude. This phenomenon is especially worse in GIN, as the node features of GIN architecture are larger due to the unnormalized sum aggregation operation (Tailor et al., 2020). Applying the proposed topology grouping with localized Wiener index further boosts the PTQ performance, as it effectively divides quantization groups in a node-wise manner, with the nodes in the group sharing similar magnitudes for the quantization.

Table 5: Inference time comparison using GCN.

| Bit | Method | Type | Reddit | | ogbn-products | |
|-----|--------|------|--------|---------|---------------|---------|
| | | | Time (s) | Speedup | Time (s) | Speedup |
| FP32 | - | - | 1.41 | - | 1.45 | - |
| INT8 | Degree-Quant | QAT | 1.22 | 1.15× | 1.30 | 1.12× |
| | SGQuant | QAT | 1.25 | 1.13× | 1.31 | 1.11× |
| | $A^2Q$ | QAT | 1.30 | 1.08× | 1.78 | 0.82× |
| | TopGQ | PTQ | 1.24 | 1.13× | 1.30 | 1.11× |

## 6.5 COST ANALYSIS

We compare the inference time in Table 5, measured on an RTX 4090 GPU with customized kernels. While the forward times are mostly similar due to the same amount of multiplications, the difference in inference time comes from the unseen vertices. While Degree-Quant does not handle unseen nodes any differently, $A^2Q$ has to perform costly nearest neighbor search on the input features. Although TopGQ performs a group search for unseen nodes, this only involves simple $I, W$ comparison before inference.

In terms of quantization time, TopGQ is orders of magnitude faster as discussed in Section 6.2. A large portion of this is due to the proposed computing method for the localized Wiener index, as shown in Table 6. We compare the time to compute the Wiener index of $k$-hop subgraph, using the same $k$ settings that are used in the main experiments. For the baselines, we used implementations from the SciPy library. The results show that our method demonstrates significant improvements in computational efficiency compared to other algorithms. Specifically, our approach reduces the Wiener index computation time by up to 602.30× on large-scale datasets like ogbn-proteins, achieving a

Table 7: Comparison on different centrality measures against localized Wiener index used in TopGQ.

| Bit | Method | PROTEINS | | | NCI1 | | |
|---|---|---|---|---|---|---|---|
| | | GCN | GIN | GraphSAGE | GCN | GIN | GraphSAGE |
| FP32 | - | 76.19 | 74.79 | 72.87 | 80.41 | 81.46 | 78.46 |
| INT4 | Degree Centrality only | 56.15 | 45.04 | 50.65 | 60.54 | 69.71 | 75.46 |
| | + Betweenness Centrality | 59.03 | 54.25 | 50.58 | 63.81 | 67.55 | 70.61 |
| | + Closeness Centrality | 58.52 | 61.73 | 50.48 | 63.14 | 69.54 | 71.97 |
| | + Katz Centrality | 53.68 | 55.24 | 44.08 | 57.19 | 57.36 | 57.77 |
| | **+ L. Wiener Index (Ours)** | 70.15 | 70.61 | 69.67 | 67.53 | 78.49 | 76.43 |

time reduction from 2.84 hours to 0.0002 hours. This trend is consistent, with speedups ranging from $9.77\times$ on Cora to $412.23\times$ on Reddit. Our method scales significantly better with larger graphs by reducing the computational cost of the Wiener index, achieving superior quantization speed.

## 6.6 ANALYSIS ON TOPOLOGICAL MEASURES

To further identify the advantages of the localized Wiener Index, we compare it against other centrality measures, such as betweenness centrality, closeness centrality, and Katz centrality. The results are shown in Table 7. The other centrality measures depict suboptimal performance compared to localized Wiener Index. We believe that the result stems from the unique expressiveness of the localized Wiener index in capturing local compactness of a node within k-hop neighbors: A small value of a node indicates a dense connectivity within its neighbors, and relatively rapid propagation of features via message passing. Therefore, TopGQ can effectively group node features with distinctive ranges, as shown in Figure 2 in the paper, leading to enhanced quantization quality.

## 6.7 ANALYSIS ON SCALE ABSORPTION

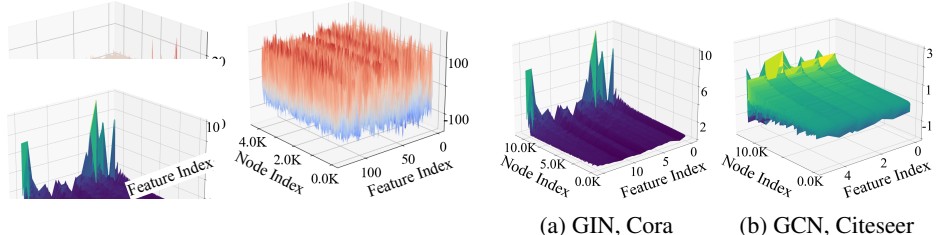

Figure 5: $X_{comb}$ before (left) and after (right) scale absorption (GCN, PROTEINS).

(a) GIN, Cora    (b) GCN, Citeseer

Figure 6: $X_{comb}$ magnitude visualization.

In this section, we visualize the activations $X_{comb}$ before and after applying scale absorption. In Figure 5, the left-hand side denotes the activation of the FP32 format before scale absorption, where visible outliers can be seen in a node-wise distribution. Such distribution with outliers results in most values being mapped to a few integers in quantization, causing inefficient use of integer precision. This is further supported by Figure 6, where the spiky distribution with large outliers is also found in other models and datasets. On the other hand, applying scale absorption leads to an even distribution across the mapped range (-128, 127) as right-hand side of Figure 5. Such even distribution depicts desirable quantization outputs of allocated integers, because the values can be mapped evenly across the bins, fully utilizing the quantization precision and thus leading to minimal quantization error.

## 7 CONCLUSION

In this paper, we propose TopGQ, the first post-training quantization method for GNNs. TopGQ proposes to group vertices that share similar topological structure, which is measured using an adaptation of Wiener index to capture the local topology around each node. For this, TopGQ proposes a new algorithm that reduces the overhead of computing localized Wiener index for each node. Then, TopGQ proposes the scale absorption method, which merges the scale parameters of quantization groups to the adjacency matrix for efficient computation. The extensive experimental results show that TopGQ outperforms baselines while having orders of magnitude faster quantization.

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

## A CODE

The code, which includes our implementation of this work, is included in a zip archive of the supplementary material. The code is under Nvidia Source Code License-NC and GNU General Public License v3.0.

## B ADDITIONAL EXPERIMENTAL SETTINGS

We report evaluation results on two representative graph processing tasks: Node-level classification, graph-level classification. For node-level classification, we compare the validation accuracy of Reddit, ogbn-proteins, and ogbn-products datasets in a transductive setting. Please note that we first conduct GNN quantization experiments on the dataset with this level of scale, thus further enlarging the field of GNN quantization. By following the experimental settings of baselines, we also conduct experiments using Cora, CiteSeer, and PubMed datasets in a transductive setting, which is the common setting for GNN quantization. Lastly, we further conduct a comparison of large-graph processing on Reddit, ogbn-proteins, and ogbn-products datasets. For graph-level classification, we choose PROTEINS and NCI1 datasets to evaluate the inductive inference performance of quantized GNNs.

We compare TopGQ with three graph quantization baselines using QAT approaches: Degree-Quant (Tailor et al., 2020), SGQuant (Feng et al., 2020), and $A^2Q$ (Zhu et al., 2022). To ensure a fair comparison, we use a fixed-precision quantization for both SGQuant and $A^2Q$ when attaining experiment results. We report quantized accuracy of GCN (Kipf & Welling, 2016), GIN (Xu et al., 2019),

Table 8: Comparison on inference and quantization time.

| Metrics | Acc. (%) | Infer. Time (s) | Infer. Speedup | Quant. Time (h) | Quant. Speedup |
|---|---|---|---|---|---|
| FP32 | 78.41 | 1.450 | $1\times$ | - | - |
| Degree-Quant | 75.26 | 1.295 | $1.120\times$ | 95.95 | $1\times$ |
| Degree-Quant-PTQ | 46.57 | 1.294 | $1.121\times$ | 0.28 | $343\times$ |
| TopGQ | 76.94 | 1.304 | $1.112\times$ | 0.34 | $282\times$ |

Table 9: Comparison of theoretical costs and storage for different methods.

| Metrics | Theoretical Cost | Theoretical Storage |
|---|---|---|
| FP32 | $O_{FP}(N^2F_1 + NF_1F_2)$ | $O_{FP}(E + F_1F_2 + NF_0)$ |
| Degree-Quant | $O_{INT}(N^2F_1 + NF_1F_2) + O_{FP_{elem}}(NF_2)$ | $O_{INT}(E + F_1F_2 + NF_0) + O_{FP}(1)$ |
| Degree-Quant-PTQ | $O_{INT}(N^2F_1 + NF_1F_2) + O_{FP_{elem}}(NF_2)$ | $O_{INT}(E + F_1F_2 + NF_0) + O_{FP}(1)$ |
| TopGQ | $O_{INT}(N^2F_1 + NF_1F_2) + O_{FP_{elem}}(NF_2)$ | $O_{INT}(E + F_1F_2 + NF_0) + O_{FP}(N_T + F_2)$ |

and GraphSAGE (Hamilton et al., 2017) architectures with 4-bit and 8-bit integer quantization. For a fair comparison, we apply the same bitwidth for all layers, including aggregation and combination.

All experiments are conducted on a server with a single A6000 GPU, RTX 4090 GPU, and Intel(R) Xeon(R) Gold 6442Y CPU. We implement our algorithm on PyG library v2.6.0 with PyTorch v2.2.1. In the Wiener index computation time comparison, we use the SciPy library to measure the time of the baseline algorithm to compute the all-pair shortest-path metric.

For the ablation study, we present in Table 4, we first build baseline PTQ method, which applies a min-max quantization strategy to quantize graph neural networks without node grouping and scale absorption. For the case of using Indegree for the node grouping metric, we apply the same strategy with our method that uses the Wiener index by grouping the nodes having the same indegree value and quantizing them to share the same quantization parameters.

## C  QUANTIZATION TRADE-OFF ANALYSIS

Here, we present a comprehensive analysis regarding the trade-offs of quantization and provide a theoretical analysis of computation cost and storage consumption. The experimental results are shown in Tables 8 and 9.

As shown in Table 8, TopGQ finds a good balance between reducing quantization time and preserving accuracy, while other choices in FP32, Degree-Quant, TopGQ demonstrate disadvantages in either accuracy, time, or memory. FP32 suffers from the expensive costs of computation and storage. While Degree-Quant alleviates this cost via quantization, the long quantization time is required to obtain the benefits. TopGQ is free from the quantization time problem but at the cost of considerable performance degradation. TopGQ aims to find the best way of addressing each issue by leveraging topological node similarities with an additional amount of storage cost.

As for the theoretical costs (Table 9), we assume GNN layer propagation as AXW operation, with $A \in \mathbb{R}^{N*N}, X \in \mathbb{R}^{N*F1}, W \in \mathbb{R}^{F1*F2}$ with initial dataset size of $N*F0$. We note computation and storage costs of FP and Int operation as follows:

- $O_{FP}()$: Complexity for floating-point operations / Storage complexity for floating-point values.

- $O_{FP_{elem}}()$: Complexity for element-wise floating-point operations.

- $O_{INT}()$: Complexity for fixed-point operations / Storage complexity for fixed-point values.

The computation cost shows that quantization converts the expensive floating-point matrix multiplication into integer operations. The additional floating-point cost comes from converting integer outputs

Table 10: Comparison of GAT architecture on citation datasets.

| Bit | Method | Type | Cora Acc. (%) | Cora Q. Time (s) | Citeseer Acc. (%) | Citeseer Q. Time (s) | PubMed Acc. (%) | PubMed Q. Time (s) |
|---|---|---|---|---|---|---|---|---|
| FP32 | - | - | 82.10 | - | 74.10 | - | 79.42 | - |
| INT4 | Degree-Quant | QAT | 80.70 | (18.78s) | 23.10 | (40.91s) | 74.50 | (60.89s) |
| | SGQuant | QAT | 74.70 | (5.55s) | 66.20 | (8.67s) | 72.40 | (9.77s) |
| | $A^2Q$ | QAT | 76.80 | (2.44s) | 61.80 | (2.47s) | 70.50 | (3.16s) |
| | TopGQ | PTQ | 80.34 | (0.92s) | 66.92 | (1.20s) | 78.06 | (1.25s) |
| INT8 | Degree-Quant | QAT | 81.70 | (18.30s) | 69.80 | (41.31s) | 79.20 | (61.02s) |
| | SGQuant | QAT | 79.90 | (5.71s) | 68.40 | (8.72s) | 76.00 | (9.74s) |
| | $A^2Q$ | QAT | 77.50 | (2.44s) | 69.50 | (2.55s) | 72.80 | (3.11s) |
| | TopGQ | PTQ | 82.02 | (0.86s) | 73.70 | (1.11s) | 79.32 | (1.26s) |

Table 11: Comparison of GAT architecture on graph classification datasets.

| Bit | Method | Type | Proteins Acc. (%) | Proteins Q. Time (s) | NCI1 Acc. (%) | NCI1 Q. Time (s) |
|---|---|---|---|---|---|---|
| FP32 | - | - | 75.56 | - | 79.73 | - |
| INT4 | Degree-Quant | QAT | 71.96 | (3626.77s) | 74.01 | (8078.41s) |
| | SGQuant | QAT | 59.56 | (267.66s) | 58.49 | (754.85s) |
| | $A^2Q$ | QAT | 70.36 | (396.62s) | 66.16 | (1002.95s) |
| | TopGQ | PTQ | 69.09 | (4.71s) | 69.70 | (9.90s) |
| INT8 | Degree-Quant | QAT | 72.41 | (3580.78s) | 74.50 | (7988.47s) |
| | SGQuant | QAT | 68.82 | (267.65s) | 74.42 | (753.09s) |
| | $A^2Q$ | QAT | 72.42 | (385.62s) | 72.28 | (997.62s) |
| | TopGQ | PTQ | 75.74 | (4.87s) | 79.48 | (9.49s) |

back to floating-point values. The measurement is provided by using our kernel, and the theoretical analysis is based on Zhu et al. (2022).

# D    GAT QUANTIZATION RESULTS

The GAT's attention-based edge weights are computed at runtime, therefore quantization scales of the adjacency matrix are also computed at runtime, meaning our method of absorbing the scale to the adjacency matrix cannot be precomputed. However, scale absorption can be modified to accommodate such dynamic quantization scenarios, which we provide in Tables 10 and 11. The results show that TopGQ also performs well in GAT architecture. In our modified scale absorption for GAT, the absorption is performed at runtime right before the quantization operation, simply by adding an FP32 element-wise multiplication between the adjacency matrix and the precalculated scales.

# E    GRAPHSAGE QUANTIZATION RESULTS WITH MEAN AGGREGATORS

We would like to present the quantization results of TopGQ for graphSAGE architecture with "mean" aggregators, with FP32 accuracies comparable to those of ogbn-leaderboard scores. In the leaderboard, the selected aggregator function is "mean", while our setting selected "max" as the aggregator function in the experiments. As presented in Table 12, TopGQ can preserve performance regardless of aggregator functions.

Table 12: Comparison of GAT architecture on graph classification datasets.

| Bit | Method | Type | Proteins | | NCI1 | |
|-----|--------|------|----------|----------|--------|----------|
| | | | Acc. (%) | Q. Time (s) | Acc. (%) | Q. Time (s) |
| FP32 | - | - | 75.56 | - | 79.73 | - |
| INT4 | Degree-Quant | QAT | 71.96 | (3626.77s) | 74.01 | (8078.41s) |
| | SGQuant | QAT | 59.56 | (267.66s) | 58.49 | (754.85s) |
| | $A^2Q$ | QAT | 70.36 | (396.62s) | 66.16 | (1002.95s) |
| | TopGQ | PTQ | 69.09 | (4.71s) | 69.70 | (9.90s) |
| INT8 | Degree-Quant | QAT | 72.41 | (3580.78s) | 74.50 | (7988.47s) |
| | SGQuant | QAT | 68.82 | (267.65s) | 74.42 | (753.09s) |
| | $A^2Q$ | QAT | 72.42 | (385.62s) | 72.28 | (997.62s) |
| | TopGQ | PTQ | 75.74 | (4.87s) | 79.48 | (9.49s) |

Table 13: Experiment results reported with standard deviation using Citation datasets.

| Bit | Model | Cora | | | Citeseer | | | Pubmed | | |
|-----|-------|------|-----|-----|----------|-----|-----|--------|-----|-----|
| | | GCN | GIN | GS | GCN | GIN | GS | GCN | GIN | GS |
| INT4 | Degree-Quant | $79.02 \pm 0.55$ | $71.88 \pm 5.10$ | $73.50 \pm 1.23$ | $22.34 \pm 1.57$ | $47.92 \pm 7.66$ | $17.14 \pm 2.96$ | $78.62 \pm 0.71$ | $76.56 \pm 10.90$ | $78.18 \pm 1.81$ |
| | SGQuant | $79.02 \pm 0.82$ | $70.21 \pm 5.22$ | $75.30 \pm 3.31$ | $68.08 \pm 0.91$ | $46.70 \pm 5.82$ | $48.34 \pm 5.93$ | $76.08 \pm 0.92$ | $65.28 \pm 7.01$ | $71.08 \pm 2.21$ |
| | $A^2Q$ | $52.68 \pm 5.82$ | $64.64 \pm 4.14$ | $74.16 \pm 0.64$ | $54.00 \pm 6.12$ | $46.04 \pm 7.75$ | $66.22 \pm 4.24$ | $69.72 \pm 4.54$ | $51.90 \pm 7.66$ | $73.92 \pm 3.84$ |
| | TopGQ | $81.50 \pm 0.44$ | $78.58 \pm 0.42$ | $79.64 \pm 0.15$ | $71.90 \pm 0.37$ | $70.14 \pm 0.34$ | $71.76 \pm 0.58$ | $79.58 \pm 0.12$ | $77.70 \pm 0.14$ | $79.00 \pm 0.16$ |
| INT8 | Degree-Quant | $81.80 \pm 0.70$ | $74.64 \pm 5.00$ | $77.50 \pm 1.09$ | $69.72 \pm 0.69$ | $58.34 \pm 7.95$ | $69.10 \pm 4.73$ | $79.24 \pm 0.78$ | $79.70 \pm 11.07$ | $78.42 \pm 1.03$ |
| | SGQuant | $80.51 \pm 0.59$ | $73.32 \pm 4.23$ | $75.32 \pm 3.86$ | $68.34 \pm 0.48$ | $51.30 \pm 5.01$ | $54.12 \pm 5.15$ | $78.06 \pm 0.54$ | $75.22 \pm 2.44$ | $73.44 \pm 0.62$ |
| | $A^2Q$ | $79.96 \pm 2.28$ | $78.74 \pm 2.68$ | $76.12 \pm 3.09$ | $70.48 \pm 1.29$ | $67.26 \pm 5.13$ | $66.04 \pm 3.04$ | $76.44 \pm 1.29$ | $76.40 \pm 0.98$ | $75.36 \pm 0.60$ |
| | TopGQ | $82.08 \pm 0.39$ | $78.42 \pm 0.53$ | $80.30 \pm 0.61$ | $72.28 \pm 0.53$ | $70.26 \pm 0.60$ | $71.96 \pm 0.75$ | $80.30 \pm 0.19$ | $78.62 \pm 0.74$ | $78.94 \pm 0.47$ |

# F  EXPERIMENT RESULTS WITH STANDARD DEVIATION IN CITATIONS DATASETS

In the main experiment tables, we omitted the error bar for better readability. To show the error range of both the baseline methods and TopGQ, we present the accuracy table with standard deviation values using citation datasets in Table 13.

# G  EXPERIMENTAL RESULTS OF QAT-STYLE TOPGQ

In neural network quantization, enabling PTQ (non-training quantization) is recognized as a contribution for two reasons: 1) PTQ is considered more efficient than QAT for practical deployment. 2) Enabling PTQ is usually difficult due to a severe accuracy drop compared to QAT. This is because PTQ has limited capacity than QAT which can freely update weights. Thus, simply building a stable PTQ method that can minimize such accuracy loss is difficult and is considered a meaningful contribution. Nevertheless, we provide TopGQ with QAT settings and compare the results with the original TopGQ, shown in Table 14 and Table 15.

We can observe that TopGQ with QAT can perform to a significant level, with several settings close to the original TopGQ. This shows that the proposed quantization techniques of TopGQ leveraging topology can also be effective in a QAT setting.

# H  LOCALIZED WIENER INDEX CALCULATION COST OF UNSEEN NODES

During inference, we only need to calculate the Wiener indices on the unseen nodes. To compare this overhead with the inference time, we provide comparison results in GCN INT8 settings with the graph datasets PROTEINS and NCI1 (Table 16). The nodes from the test set graphs will be the unseen nodes, and their information will have to be calculated during inference time.

Table 14: Comparing QAT and PTQ implementations of TopGQ on node classification task.

| Method | Bit | Cora | | | Citeseer | | | Pubmed | | |
| --- | --- | --- | --- | --- | --- | --- | --- | --- | --- | --- |
| | | GCN | GIN | GraphSAGE | GCN | GIN | GraphSAGE | GCN | GIN | GraphSAGE |
| TopGQ + QAT | INT4 | 80.08 | 76.30 | 76.64 | 70.58 | 69.10 | 69.56 | 78.50 | 77.00 | 76.72 |
| TopGQ | | 81.50 | 78.58 | 79.64 | 71.90 | 70.14 | 71.76 | 79.58 | 77.70 | 79.00 |
| TopGQ + QAT | INT8 | 81.12 | 78.30 | 76.00 | 70.24 | 69.14 | 69.50 | 79.40 | 78.86 | 78.10 |
| TopGQ | | 82.08 | 78.42 | 80.30 | 72.28 | 70.26 | 71.96 | 80.30 | 78.62 | 78.94 |

Table 15: Comparing QAT and PTQ implementations of TopGQ on graph classification task.

| Method | Bit | PROTEINS | | | NCI1 | | |
| --- | --- | --- | --- | --- | --- | --- | --- |
| | | GCN | GIN | GraphSAGE | GCN | GIN | GraphSAGE |
| TopGQ + QAT | INT4 | 67.36 | 66.31 | 66.61 | 69.74 | 66.66 | 73.12 |
| TopGQ | | 69.94 | 70.92 | 68.93 | 65.88 | 75.37 | 75.98 |
| TopGQ + QAT | INT8 | 73.19 | 65.68 | 73.73 | 77.86 | 79.86 | 77.92 |
| TopGQ | | 75.65 | 74.34 | 72.20 | 79.72 | 81.36 | 78.43 |

As we can see in the table, the overhead for calculating Wiener index of unseen test nodes accounts for only a very small portion (smaller than 1%) of the total inference time. Note that this is made possible from a specialized algorithm to accelerate the computation of the localized Wiener Index, which is another contribution of TopGQ.

# I  ANALYSIS OF TOPOLOGY-AWARE GROUPING ON PTQ AND QAT

As we discussed in the main body of the paper, the better performance of TopGQ compared to QAT baselines is from better consideration of topological structure. In other words, the superior performance of TopGQ is due to a better ability to find quantization parameters, and is orthogonal to the PTQ/QAT differences. To validate this, we present two variants: Degree-Quant-PTQ and TopGQ-QAT, which are the PTQ and QAT versions of each method, respectively. The experimental results are shown in Table 17. The results show that our proposed topology-aware grouping shows better performance regardless of PTQ and QAT.

As for the results that outperform FP32 accuracies, we believe this phenomenon often occurs when the low-bit format is sufficient to handle the original model complexity. We cite some papers that exhibit the mentioned occasion in their experiments. For example, some of Wu et al.; Shomron et al. (2021) report better performance in 8-bit settings than FP32 settings at various tasks.

# J  FURTHER COMPARISON ON $k$-HOP WIENER INDEX

Here, we present further analysis of the Wiener index, including a sensitivity study regarding the hyperparameter $k$, which determines the diameter of the local subgraph.

Tables 18 and 20 shows the sensitivity study on quantization accuracy regarding the hop size $k$. We compare the hop size $k = 1$, $k = 2$, and $k = 3$, where the $k = 1$ setting corresponds to the baseline,

Table 16: Average inference time and calculation overhead for GCN on PROTEINS and NCI1 datasets.

| hop size $k$ | $k = 2$ | | $k = 3$ | |
| --- | --- | --- | --- | --- |
| Metric | PROTEINS | NCI1 | PROTEINS | NCI1 |
| Avg. test inference time (s) | 0.0438 | 0.1332 | 0.0438 | 0.1332 |
| Avg. overhead of calculation (s) | 0.0001 | 0.0003 | 0.0018 | 0.0048 |
| Proportion | 0.23% | 0.22% | 4.07% | 3.63% |

Table 17: Comparison on PTQ and QAT differences on INT4 Quantization.

| Method | Cora | | | PubMed | | |
|---|---|---|---|---|---|---|
| | GCN | GIN | GraphSAGE | GCN | GIN | GraphSAGE |
| Degree-Quant | 79.00 | 71.90 | 73.50 | 78.60 | 76.60 | 78.20 |
| Degree-Quant-PTQ | 78.42 | 30.46 | 78.54 | 78.34 | 50.20 | 77.64 |
| TopGQ-QAT | 80.08 | 76.30 | 76.64 | 78.50 | 77.00 | 76.72 |
| TopGQ | 81.50 | 78.58 | 79.64 | 79.58 | 77.70 | 79.00 |

Table 18: Sensitivity study of TopGQ on graph classification task.

| Bit | Datasets | PROTEINS | | | NCI1 | | |
|---|---|---|---|---|---|---|---|
| | Hop size $k$ | GCN | GIN | GraphSAGE | GCN | GIN | GraphSAGE |
| INT4 | $k=1$ | 73.34 | 72.88 | 73.03 | 80.81 | 81.60 | 78.88 |
| | $k=2$ | 76.05 | 74.61 | 74.22 | 80.86 | 81.84 | 79.10 |
| | $k=3$ | 75.94 | 74.86 | 74.00 | 80.91 | 81.88 | 79.16 |
| INT8 | $k=1$ | 60.86 | 51.04 | 65.77 | 62.68 | 70.91 | 75.90 |
| | $k=2$ | 66.06 | 63.96 | 67.01 | 66.14 | 77.33 | 76.50 |
| | $k=3$ | 70.15 | 70.61 | 69.67 | 65.09 | 78.49 | 76.43 |

Table 19: Comparison of node-wise Wiener index computation time.

| $k$ | Algorithm | Cora | CiteSeer | Pubmed | PROTEINS | NCI1 | Reddit | ogbn-proteins | ogbn-products |
|---|---|---|---|---|---|---|---|---|---|
| 2 | Bellman-Ford | 0.35s | 0.46s | 19.87s | 13.02s | 33.01s | 4.21h | 2.84h | 34.64h |
| | Floyd-Warshall | 0.15s | 0.21s | 4.68s | 7.91s | 2.97s | 0.57h | 0.41h | 4.51h |
| | Dijkstra | 0.19s | 0.29s | 2.70s | 8.78s | 2.56s | 0.16h | 0.11h | 1.55h |
| | Ours | 0.02s | 0.01s | 0.06s | 0.29s | 0.10s | 0.0004h | 0.0002h | 0.0048h |
| | Speed Up | 9.77× | 31.65× | 43.50× | 27.12× | 25.17× | 412.23× | 602.30× | 322.37× |
| 3 | Bellman-Ford | 3.63s | 2.43s | 216.76s | 40.07s | 512.13s | 46.63h | 30.09h | 305.50h |
| | Floyd-Warshall | 0.77s | 0.31s | 34.62s | 8.50s | 11.27s | 5.75h | 3.78h | 35.44h |
| | Dijkstra | 0.61s | 0.39s | 14.71s | 12.75s | 12.49s | 1.02h | 0.61h | 8.52h |
| | Ours | 0.52s | 0.08s | 1.82s | 4.84s | 1.78s | 0.0155h | 0.0065h | 0.2855h |
| | Speed Up | 1.18× | 3.76× | 8.10× | 1.76× | 6.32× | 65.89× | 93.39× | 29.83× |

Table 20: Sensitivity study of TopGQ on citation datasets.

| Bit | Datasets | Cora | | | Citeseer | | | PubMed | | |
|---|---|---|---|---|---|---|---|---|---|---|
| | Hop size $k$ | GCN | GIN | GraphSAGE | GCN | GIN | GraphSAGE | GCN | GIN | GraphSAGE |
| INT4 | $k=1$ | 81.00 | 77.12 | 78.82 | 71.86 | 69.10 | 70.86 | 78.20 | 75.42 | 78.50 |
| | $k=2$ | 81.50 | 78.58 | 79.64 | 71.90 | 70.14 | 71.76 | 79.58 | 77.70 | 79.00 |
| | $k=3$ | 81.56 | 78.32 | 78.56 | 72.12 | 70.47 | 71.38 | 79.20 | 77.00 | 78.72 |
| INT8 | $k=1$ | 81.96 | 78.36 | 79.92 | 72.24 | 70.18 | 71.84 | 80.18 | 78.34 | 78.90 |
| | $k=2$ | 82.08 | 78.42 | 80.30 | 72.28 | 70.26 | 71.96 | 80.30 | 78.62 | 78.94 |
| | $k=3$ | 82.10 | 78.38 | 79.54 | 72.24 | 70.60 | 71.92 | 80.24 | 78.68 | 78.84 |

Table 21: Comparison on different centrality measures against localized Wiener index used in TopGQ.

| Bit | Method | PROTEINS | | | NCI1 | | |
|---|---|---|---|---|---|---|---|
| | | GCN | GIN | GraphSAGE | GCN | GIN | GraphSAGE |
| FP32 | - | 76.19 | 74.79 | 72.87 | 80.41 | 81.46 | 78.46 |
| INT4 | Degree Centrality only | 56.15 | 45.04 | 50.65 | 60.54 | 69.71 | 75.46 |
| | + Betweeness Centrality | 59.03 | 54.25 | 50.58 | 63.81 | 67.55 | 70.61 |
| | + Closeness Centrality | 58.52 | 61.73 | 50.48 | 63.14 | 69.54 | 71.97 |
| | + Katz Centrality | 53.68 | 55.24 | 44.08 | 57.19 | 57.36 | 57.77 |
| | **+ L. Wiener Index (Ours)** | 70.15 | 70.61 | 69.67 | 67.53 | 78.49 | 76.43 |
| INT8 | Degree Centrality only | 72.57 | 71.86 | 70.48 | 78.91 | 81.28 | 78.32 |
| | + Betweeness Centrality | 62.10 | 61.55 | 55.08 | 76.89 | 75.18 | 75.13 |
| | + Closeness Centrality | 62.48 | 64.96 | 57.33 | 76.49 | 76.68 | 75.85 |
| | + Katz Centrality | 56.82 | 57.97 | 48.56 | 64.20 | 62.19 | 64.27 |
| | **+ L. Wiener Index (Ours)** | 75.94 | 74.86 | 74.00 | 80.91 | 81.88 | 79.16 |

which is identical to the "indegree" setting in Table 4. For the PROTEINS dataset, increasing the hop size from $k = 1$ to $k = 2$ led to noticeable improvements across all models.

Across both precision levels and datasets, a clear trend emerged where increasing the hop size from $k = 1$ to $k = 2$ generally improved performance for all architectures. This effect was particularly noticeable in the PROTEINS dataset under INT4 precision, where all architectures showed consistent gains. In the INT8 configuration, the same trend held, though the magnitude of improvements was more pronounced in some cases. Notably, GIN and GCN showed substantial increases in performance as the hop size increased from $k = 1$ to $k = 3$ for the PROTEINS dataset, while the NCI1 dataset saw more moderate gains. However, increasing the hop size further to $k = 3$ did not always lead to continued improvements.

We also conducted further sensitivity study regarding Wiener index computation time by varying the value of $k$, as increasing $k$ results in more computational costs due to the larger diameters of each subgraph. Table 19 shows the comparison results of computation time.

For $k = 2$, our method shows remarkable speedups, often outperforming the other algorithms by significant margins, especially for larger graphs where it achieves up to several hundred times faster performance. At $k = 3$, while all methods take longer due to the increased complexity, our method continues to lead in performance, though the speedup is generally lower than for $k = 2$. Nonetheless, it maintains a strong advantage, especially in large-scale cases where traditional methods struggle with execution times. Overall, the trends show that our method provides consistent and substantial speed improvements.

From the experiments, we observe that $k = 2$ often provides the best balances of quantization accuracy and computation time. In addition, for better quantization accuracy, using $k = 3$ is also an option to choose.

## K    COMPARISON AGAINST DIFFERENT CENTRALITY MEASURES

In this section, we provide quantization accuracies when using other centrality measures than our proposed localized Wiener index. Table 21 shows the results using PROTEINS and NCI1 datasets on three different model architectures, both in INT4 and INT8 precisions. As we observed in Section 6.6, the results are consistent in that localized Wiener index shows superior results to other centrality measures.

