# OpenReview forum: "TopGQ: Post-Training Quantization for GNNs via Topology Based Node Grouping"
_ICLR.cc/2025/Conference — Submitted to ICLR 2025_

### Official Review · Reviewer_fkMV · 2024-10-29

**Soundness:** 2
**Presentation:** 1
**Contribution:** 2
**Rating:** 5
**Confidence:** 3

**Summary:**

TopGQ proposes a post-training quantization (PTQ) framework for Graph Neural Networks (GNNs), achieving favorable quantization results without the need for backpropagation. To address the challenge of significant diversity in node features, TopGQ introduces a local Wiener index from a grouping perspective, clustering nodes with similar in-degrees and local Wiener indices for quantization. Additionally, TopGQ optimizes the computation of the local Wiener index, enhancing the efficiency of the grouping process. Finally, it employs scale absorption techniques to merge feature scales into the adjacency matrix, resulting in a more uniform feature distribution.

I read the rebuttals and decided to keep my decision. In my eyes, adding too many experiments in the rebuttal should not be encouraged. Besides, this paper still lacks comprehensive discussions on GNN quantization.

**Strengths:**

1. A framework for GNN post-training quantization (PTQ) without backpropagation is proposed, achieving superior quantization results with reduced calibration time.
2. The concept of grouping is introduced, utilizing the Wiener index as a replacement for the traditional in-degree metric for effective grouping.
3. An accelerated algorithm for computing the Wiener index is presented, enhancing parallelism and reducing overhead during the grouping process.
4. The code is released, and practical deployment is demonstrated on a GPU.

**Weaknesses:**

1. The literature review is insufficient. Many SOTA works [1-3] are not mentioned and have not been included in the experimental comparisons. In the summary of GNN quantization works, only Degree-Quant, SgQuant, and A2Q (highlighted in red) were compared in the experiments, lacking consideration of newer works.
2. Section 5.1: The rationale for using the Wiener index as the basis for grouping is not explained. As stated in the paper, there are various metrics to describe the topology of graphs. More theoretical derivation or comparative experiments are needed to demonstrate the advantages of using the Wiener index over other metrics.
3. Section 5.3: The introduction of scale absorption does not clarify its purpose. According to Equation (16), it merges the scale of X into the adjacency matrix A after quantization. However, the results provided in Section 6.6 show that the distribution before quantization is more uniform and is considered easier for quantization, which appears inconsistent with the method presented in Section 5.3.
4. Lack of Training and Without-Training Comparative Experiments: Without-training should only be considered a viable option when training performance is average or shows minimal improvement. It is inappropriate to directly present without-training as a contribution, as it is relatively easy to implement.
5. Section 6.4: The use of Scale Absorption for INT8 generally results in precision loss, which is not explained.
6. Section 6.5: Although actual inference speedup is provided, INT8 only achieves 1.25 times acceleration compared to FP32, or even lower. However, reference [2] demonstrates 3-4 times or higher inference speedup. Additionally, while INT4 is mentioned, there is no deployment of INT4, requiring a reasonable explanation.
7. Section 6.5: The presented speed performance of the optimized local Wiener index is compared to very outdated classical methods, which weakens the argument's credibility.

[1] EPQuant: A Graph Neural Network compression approach based on product quantization [NC 2022]

[2] Low-bit Quantization for Deep Graph Neural Networks with Smoothness-aware Message Propagation [CIKM 2023]

[3] Haar wavelet feature compression for quantized graph convolutional networks [TNNLS 2023]

**Questions:**

1. This paper states that TopGQ is the first PTQ framework for GNNs; however, according to the review [1], SGQuant is actually a PTQ work. The paper also claims that SGQuant is a method for Quantization-Aware Training (QAT), which requires verification.
2. The experimental results of [2] and A2Q [3] exhibit fluctuations (e.g., 81.5±0.7%), while the experimental results presented in this paper are fixed values. Is this a normal phenomenon?

[1] A Survey on Graph Neural Network Acceleration: Algorithms, Systems, and Customized Hardware

[2] Low-bit Quantization for Deep Graph Neural Networks with Smoothness-aware Message Propagation [CIKM 2023]

[3] A2Q: Aggregation-Aware Quantization for Graph Neural Networks [ICLR 2022]

---

> ### Author Response · Authors · 2024-11-22
>
> ### **W1. The literature review is insufficient. Many SOTA works are not mentioned and have not been included in the experimental comparisons. In the summary of GNN quantization works, only Degree-Quant, SGQuant, and $A^2Q$ (highlighted in red) were compared in the experiments, lacking consideration of newer works.**
>
> $\to$ We acknowledged the weakness in the literature review of GNN quantization, and updated the recommended papers in the revised version of TopGQ. If the reviewer has additional feedbacks regarding the updated version, please let us know.
>
> We are currently working on implementing the recommended SOTA works with our evaluation architecture to enhance our paper. We will promptly let the reviewer know as soon as the tables are ready to report.
>
> ### **W2. Section 5.1: The rationale for using the Wiener index as the basis for grouping is not explained. As stated in the paper, there are various metrics to describe the topology of graphs. More theoretical derivation or comparative experiments are needed to demonstrate the advantages of using the Wiener index over other metrics.**
>
> $\to$ To further identify the advantages that localized Wiener Index has for quantization, we compared the quantization results with Proteins and NCI1 datasets over other graph properties such as betweenness centrality, closeness centrality, and Katz centrality. The results are shown below.
>
> The other node centrality measures depict suboptimal performance compared to using localized Wiener Index, in both INT4 and INT8 settings.
> We believe that the result stems from the unique expressiveness of the localized Wiener Index in capturing local compactness of a node within k-hop neighbors: A small value of a node indicates a dense connectivity within its neighbors, and relatively rapid propagation of features via message passing. Therefore, TopGQ can effectively group node features with distinctive ranges, as shown in Figure 2 in the paper, leading to enhanced quantization quality.
>
> |Method|Bit|Proteins GCN|Proteins GIN|Proteins GS|NCI1 GCN|NCI1 GIN|NCI1 GS|
> |-|-|-:|-:|-:|-:|-:|-:|
> |-| FP32 |76.19|74.79|72.87|80.41|81.46|78.46|
> |
> |Degree Centrality only|INT8|72.57|71.86|70.48|78.91|81.28|78.32|
> |+ Betweenness Centrality|INT8|62.10|61.55|55.08|76.89|75.18|75.13|
> |+ Closeness Centrality|INT8|62.48|64.96|57.33|76.49|76.68|75.85|
> |+ Katz Centrality|INT8|56.82|57.97|48.56|64.20|62.19|64.27|
> |+ Ours|INT8|75.94|74.86|74.00|80.91|81.88|79.16|
> |
> |Degree Centrality only|INT4|56.15|45.04|50.65|60.54|69.71|75.46|
> |+ Betweenness Centrality|INT4|59.03|54.25|50.58|63.81|67.55|70.61|
> |+ Closeness Centrality|INT4|58.52|61.73|50.48|63.14|69.54|71.97|
> |+ Katz Centrality|INT4|53.68|55.24|44.08|57.19|57.36|57.77|
> |+ Ours|INT4|70.15|70.61|69.67|67.53|78.49|76.43|
>
> ### **W3. Section 5.3: The introduction of scale absorption does not clarify its purpose. According to Equation (16), it merges the scale of X into the adjacency matrix A after quantization. However, the results provided in Section 6.6 show that the distribution before quantization is more uniform and is considered easier for quantization, which appears inconsistent with the method presented in Section 5.3.**
>
> $\to$ The purpose of Scale Absorption is to preserve integer-format aggregation speedups while mitigating quantization challenges in GNN activations. GNNs have quantization difficulties induced by the nature of activations in GNN layers. Activation outliers occur node-wise due to the message-passing mechanism in GNN layers, where repeated aggregation can amplify values, leading to significant outliers. This observation is illustrated in Figure 5. Its application prevents activations from quantizing activations in a poor feature-wise(column-wise) manner, and ensures integer operations in aggregation.
>
> We have acknowledged the clarity issue for the readers, and revised the section of Scale Absorption and its analysis in Section 6.7, to further clarify its design objective. If it needs more improvements, we will be happy to hear in the discussion, please let us know.
>
> As to any confusion by our description, we also provide an additional explanation of Figure 5 mentioned in Section 6.7 in the comments. The left figure shows the FP32 format of $X_{comb}$, while the right figure depicts its quantized INT8 version after applying scale absorption. In the FP32 representation, significant node-wise outliers are visible. When it is quantized in a feature-wise manner (i.e., activations with the same feature index are quantized together), it results in most values being mapped to a few integers by outliers, causing a skewed distribution and inefficient use of integer range (bits).
>
> Scale absorption addresses this issue by enabling node-wise quantization, which isolates node-wise outliers for separate quantization. It ensures more evenly distributed values across the integer range, as seen in the right figure, where the distribution of quantized values appears significantly uniform.

---

> ### Author Response · Authors · 2024-11-22
>
> ### **W4. Lack of Training and Without-Training Comparative Experiments: Without-training should only be considered a viable option when training performance is average or shows minimal improvement. It is inappropriate to directly present without training as a contribution, as it is relatively easy to implement.**
>
> $\to$ In neural network quantization, enabling PTQ (non-training quantization) is recognized as a contribution for two reasons: 1) PTQ is considered more efficient than QAT for practical deployment. 2) Enabling PTQ is usually difficult due to a severe accuracy drop compared to QAT.
> This is because PTQ has limited capacity than QAT which can freely update weights. Thus, simply building a stable PTQ method that can minimize such accuracy loss is difficult and is considered a meaningful contribution.
>
> Nevertheless, we agree with the reviewer that comparing with and without-training can further enhance the soundness of our paper. Thus we provide TopGQ with QAT settings and compare the results with the original TopGQ.
> We can observe that TopGQ with QAT can perform to a significant level, with several settings close to the original TopGQ. We thank the reviewer for the feedback on the experiment suggestion, and for the discovery that the proposed quantization techniques of TopGQ leveraging topology can also be effective in a QAT setting.
>
> |Method|Bit|Cora|GCN|GIN|GS|Citeseer|GCN|GIN|GS|Pubmed|GCN|GIN|GS|
> |-|:-:|-|-:|-:|-:|-|-:|-:|-:|-|-:|-:|-:|
> |TopGQ + QAT|INT8||81.12|78.30|76.00||70.24|69.14|69.50||79.40|78.86|78.10|
> |Original TopGQ|INT8||82.08|78.42|80.30||72.28|70.26|71.96||80.30|78.62|78.94|
> |
> |TopGQ + QAT|INT4||80.08|76.30|76.64||70.58|69.10|69.56||78.50|77.00|76.72|
> |Original TopGQ|INT4||81.50|78.58|79.64||71.90|70.14|71.76||79.58|77.70|79.00|
>
>
> |Model|Bit|PROTEINS|GCN|GIN|GS|NCI1|GCN|GIN|GS|
> |-|:-:|-|-:|-:|-:|-|-:|-:|-:|
> |TopGQ + QAT|INT8||73.19|65.68|73.73||77.86|79.86|77.92|
> |Original TopGQ|INT8||75.65|74.34|72.20||79.72|81.36|78.43|
> |
> |TopGQ + QAT|INT4||67.36|66.31|66.61||69.74|66.66|73.12|
> |Original TopGQ|INT4||69.94|70.92|68.93||65.88|75.37|75.98|
>
>
> ### **W5. Section 6.4: The use of Scale Absorption for INT8 generally results in precision loss, which is not explained.**
>
> $\to$ The reason for the precision loss is because Scale Absorption quantizes FP32 quantization scale of $X$ ($s_X$) for enhancing the quality of quantized $X$. In quantizing $A \cdot X$ operation in the GNN layer, using Scale Absorption improves the precision of $X$ by absorbing its FP32 quantization scales ($s_X$) into $A$, and then quantize $A$. This indicates that enhancement of precision in activation $X$ comes at the cost of quantizing $s_X$ with $A$, potentially resulting in inaccurate scale parameters.
>
> In INT4 settings, Scale Absorption improves quantized networks as the limited range of 4bit quantization makes feature-wise quantization highly vulnerable to outliers. In INT8 settings, however, the broader integer range may allow preserving information even with outlier nodes, thus sometimes making Scale Absorption less advantageous due to scale compression effects.

---

> ### Author Response · Authors · 2024-11-23
>
> ### **W7. Section 6.5: The presented speed performance of the optimized local Wiener index is compared to very outdated classical methods, which weakens the argument's credibility.**
>
> $\to$ We are working on the additional experiment to compare our accelerated Wiener index algorithm against parallel versions of other traditional algorithms. We will promptly notify the reviewer with the updated results as soon as they are available.
>
> ### **Q1. This paper states that TopGQ is the first PTQ framework for GNNs; however, according to the review [1], SGQuant is actually a PTQ work. The paper also claims that SGQuant is a method for Quantization-Aware Training (QAT), which requires verification.**
>
> $\to$ SGQuant is a method for QAT, according to its paper. The authors of SGQuant explicitly present “GNN quantization finetuning” as a key contribution, detailed in Section III.B. This section describes settings like SGQuant using the original GNN loss and employing gradient backpropagation via STE. Although the framework of SGQuant is able to separate the training process and its methodology to adjust to the setting of PTQ, this does not mean SGQuant is strictly a PTQ-targeted method, as the original SGQuant clearly includes its training process in its proposal.
>
> In addition, several papers acknowledge SGQuant as a QAT method, which we cite below:
>
> [1] Novkin, Rodion, Florian Klemme, and Hussam Amrouch. "Approximation-and Quantization-Aware Training for Graph Neural Networks." IEEE Transactions on Computers (2023).
> [2] Ma, Yuxin, et al. "Eliminating Data Processing Bottlenecks in GNN Training over Large Graphs via Two-level Feature Compression." Proceedings of the VLDB Endowment 17.11 (2024): 2854-2866.
> [3] Tao, Zhuofu, et al. "Lw-gcn: A lightweight fpga-based graph convolutional network accelerator." ACM Transactions on Reconfigurable Technology and Systems 16.1 (2022): 1-19.
> [4] Wu, Chen, et al. "Skeletongcn: a simple yet effective accelerator for gcn training." 2022 32nd International Conference on Field-Programmable Logic and Applications (FPL). IEEE, 2022.
> [5] Saad, Leila Ben, and Baltasar Beferull-Lozano. "Quantization in graph convolutional neural networks." 2021 29th European Signal Processing Conference (EUSIPCO). IEEE, 2021.
>
> In conclusion, we would like to highlight that TopGQ is the first work to provide a GNN quantization tailored for post-training application, satisfying the traditional PTQ definition of not updating the weights through backpropagation.
>
> ### **Q2. The experimental results of [2] and $A^2Q$ [3] exhibit fluctuations (e.g., 81.5±0.7%), while the experimental results presented in this paper are fixed values. Is this a normal phenomenon?**
>
> $\to$ We observed the same fluctuations when running $A^2Q$ in our experiments, but we omitted the error bar from the main table for better readability. We present the accuracy table with standard deviation values of citation datasets as below. We will add additional results with error bars in the supplementary section F.
>
> |Method|Bit|Method|Cora|||Citeseer|||Pubmed|||
> |-----------------|------------------|-----------------|---------------------|------------------|------------------|---------------------|------------------|------------------|---------------------|------------------|------------------|
> |||**Model**|GCN|GIN|GS|GCN|GIN|GS|GCN|GIN|GS|
> |Degree-Quant|INT4||79.02$\pm$0.55|71.88$\pm$5.10|73.50$\pm$1.23|22.34$\pm$1.57|47.92$\pm$7.66|17.14$\pm$2.96|78.62$\pm$0.71|76.56$\pm$10.90|78.18$\pm$1.81|
> |SGQuant|INT4||79.02$\pm$0.82|70.21$\pm$5.22|75.30$\pm$3.31|68.08$\pm$0.91|46.70$\pm$5.82|48.34$\pm$5.93|76.08$\pm$0.92|65.28$\pm$7.01|71.08$\pm$2.21|
> |$A^2Q$|INT4||52.68$\pm$5.82|64.64$\pm$4.14|74.16$\pm$0.64|54.00$\pm$6.12|46.04$\pm$7.75|66.22$\pm$4.24|69.72$\pm$4.54|51.90$\pm$7.66|73.92$\pm$3.84|
> |TopGQ|INT4||81.50$\pm$0.44|78.58$\pm$0.42|79.64$\pm$0.15|71.90$\pm$0.37|70.14$\pm$0.34|71.76$\pm$0.58|79.58$\pm$0.12|77.70$\pm$0.14|79.00$\pm$0.16|
> |Degree-Quant|INT8||81.80$\pm$0.70|74.64$\pm$5.00|77.50$\pm$1.09|69.72$\pm$0.69|58.34$\pm$7.95|69.10$\pm$4.73|79.24$\pm$0.78|79.70$\pm$11.07|78.42$\pm$1.03|
> |SGQuant|INT8||80.51$\pm$0.59|73.32$\pm$4.23|75.32$\pm$3.86|68.34$\pm$0.48|51.30$\pm$5.01|54.12$\pm$5.15|78.06$\pm$0.54|75.22$\pm$2.44|73.44$\pm$0.62|
> |$A^2Q$|INT8||79.96$\pm$2.28|78.74$\pm$2.68|76.12$\pm$3.09|70.48$\pm$1.29|67.26$\pm$5.13|66.04$\pm$3.04|76.44$\pm$1.29|76.40$\pm$0.98|75.36$\pm$0.60|
> |TopGQ|INT8||82.08$\pm$0.39|78.42$\pm$0.53|80.30$\pm$0.61|72.28$\pm$0.53|70.26$\pm$0.60|71.96$\pm$0.75|80.30$\pm$0.19|78.62$\pm$0.74|78.94$\pm$0.47|

---

> ### Author Response · Authors · 2024-11-23
>
> We apologize for the inconvenience of our response to W6 unintentionally being omitted from the order of the previous comments. We provide it below.
>
> ### **W6. Section 6.5: Although actual inference speedup is provided, INT8 only achieves 1.25 times acceleration compared to FP32, or even lower. However, reference [2] demonstrates 3-4 times or higher inference speedup. Additionally, while INT4 is mentioned, there is no deployment of INT4, requiring a reasonable explanation.**
>
> $\to$ As the computational cost of our method and the referenced paper [1] is similar, we believe that the speedup of [1] comes from a highly optimized CUDA kernel from [2]. One of our baselines, Degree-Quant [3], uses the same per-tensor quantization as [1] but reports a 1.1× speedup compared to FP32, which is comparably lower than reported in [1]. This varying speedup despite similar quantization settings is due to the use of different CUDA kernels, not from the efficiency of the algorithms. Building a highly optimized kernel for GNN inference is another line of work that is orthogonal to ours. For now, we focus on reducing accuracy drop of GNN PTQ.
>
> Accelerated INT4 deployment on GNNs is currently very limited due to the absence of publicly available INT4 sparse matrix multiplication (SPMM) kernels that accommodate widely-used quantization settings with full support of TensorCore. Application of [2] is restricted to naive per-tensor symmetric quantization and lacks compatibility with other quantization methods. We anticipate that future kernels leveraging INT4 operations will better support efficient GNN inference.
>
> Nevertheless, Table 5 of TopGQ shows that TopGQ can accelerate inference in integer formats when paired with appropriate kernels supporting integer operations. Additionally, we improved our inference kernel to resolve speed concerns, and therefore provide a new table that presents enhanced full-batch inference time. This updated table is now included as Table 5 in the revised version of TopGQ.
>
>
> |Method|Type|Bit|Reddit (s)|Speedup|OGBN-Products (s)|Speedup|
> |-|-|-|-:|-:|-:|-:|
> |-|-|FP32|1.41|-|1.45|-|
> |Degree-Quant|QAT|INT8|1.22|1.15$\times$|1.30|1.12$\times$|
> |A2Q|QAT|INT8|1.30|1.08$\times$|1.78|0.82$\times$|
> |SGQuant|QAT|INT8|1.25|1.13$\times$|1.31|1.11$\times$|
> |TopGQ|PTQ|INT8|1.24|1.13$\times$|1.30|1.11$\times$|
>
>
> [1] Wang, Shuang, et al. "Low-bit quantization for deep graph neural networks with smoothness-aware message propagation." Proceedings of the 32nd ACM International Conference on Information and Knowledge Management. 2023.
>
> [2] Wang, Yuke, Boyuan Feng, and Yufei Ding. "QGTC: accelerating quantized graph neural networks via GPU tensor core." Proceedings of the 27th ACM SIGPLAN symposium on principles and practice of parallel programming. 2022.
>
> [3] Tailor, Shyam Anil, Javier Fernandez-Marques, and Nicholas Donald Lane. "Degree-Quant: Quantization-Aware Training for Graph Neural Networks." International Conference on Learning Representations. 2020.

---

> ### Author Response · Authors · 2024-11-26
>
> As our experiment results are ready, we provide our responses to W1 as below.
> ### **W1. The literature review is insufficient. Many SOTA works are not mentioned and have not been included in the experimental comparisons. In the summary of GNN quantization works, only Degree-Quant, SgQuant, and A2Q (highlighted in red) were compared in the experiments, lacking consideration of newer works.**
>
> $\to$ We compare EPQuant[1] and SMP[2] with citation datasets, GCN architecture.
>
> ||| **Cora** || **Citeseer** || **Pubmed**   ||
> |----------------|-----------|--------|-------------|--------|-------------|--------|-------------|
> | **Method**         | **Bit**   | **Acc.**   | **Q. Time (s)** | **Acc.**   | **Q. Time (s)** | **Acc.**   | **Q. Time (s)** |
> | **FP32**       |           | 82.08% | -           | 72.34% | -           | 80.32% | -           |
> | [1] EPQuant    | INT8      | 79.87% | 24.07       | 69.39% | 44.06       | 76.46% | 97.09       |
> | [2] SMP        | INT8      | 81.93% | 28.08       | 69.11% | 32.91       | 80.73% | 49.73       |
> | **TopGQ**  | INT8      | 82.08% | 1.12        | 72.28% | 1.11        | 80.30% | 1.08        |
> | [1] EPQuant    | INT4      | 75.62% | 24.07       | 66.41% | 44.13       | 54.99% | 97.25       |
> | [2] SMP        | INT4      | 79.33% | 28.26       | 68.00% | 34.75       | 78.67% | 49.76       |
> | **TopGQ**  | INT4      | 81.50% | 1.40        | 71.90% | 1.17        | 79.58% | 1.21        |
>
> TopGQ outperforms in accuracy with the quickest quantization time when compared to [1] and [2]. In [1], product quantization is used to compress datasets for reduced memory usage, which accounts for most of the initial quantization time. [2] introduces skewness-aware bitwise truncation and learnable ranges, which require additional computations during feature propagation in model training, resulting in longer training times.
>
> As for [3], [3] leverages compressed graph wavelet transform convolution combined with convolution layers for quantization. Due to the complexity and theoretical nature of the method described in the paper, as well as the unavailability of its implementation code, a direct and fair comparison at this stage remains challenging. Thus we aim to compare with [3] in future work.
>
> [1] Huang, Linyong, et al. "EPQuant: A Graph Neural Network compression approach based on product quantization." Neurocomputing 503, 2022.
>
> [2] Wang, Shuang, et al. "Low-bit quantization for deep graph neural networks with smoothness-aware message propagation." Proceedings of the 32nd ACM International Conference on Information and Knowledge Management. 2023.
>
> [3] Eliasof, Moshe, Benjamin J. Bodner, and Eran Treister. "Haar wavelet feature compression for quantized graph convolutional networks." IEEE Transactions on Neural Networks and Learning Systems, 2023.

---

> ### Author Response · Authors · 2024-11-26
>
> As our experiment results are ready, we provide our responses to W7 as below.
> ### **W7. Section 6.5: The presented speed performance of the optimized local Wiener index is compared to very outdated classical methods, which weakens the argument's credibility.**
>
>
> $\to$ We update the baselines with parallel algorithms on GPU for all-pair shortest paths and present its speed performance with a new table.
>
> | Datasets                | Reddit  | ogbn-proteins | ogbn-products |
> |-----------------------|---------|---------------|---------------|
> | **Method**                | **Process Time (h)**|||
> | Dijkstra              | 0.16    | 0.11          | 8.52          |
> | Parallel Dijkstra     | 0.18    | 0.13          | 6.99          |
> | Floyd-Warshall        | 0.57     | 0.41          | 35.44         |
> | Parallel Floyd-Warshall | 0.26   | 0.12          | 1.98          |
> | Ours              | 0.0004  | 0.0002        | 0.2855        |
> | Speedup               | 409.67  | 602.30        | 6.93          |
>
> In ogbn-products, both the parallel Dijkstra and parallel Floyd-Warshall methods improve the speed of the traditional approaches but remain slower than our Accelerated Localized Wiener Index computation. This is due to the proposed algorithm, where the distance between any arbitrary node pair is limited by a hop-count of k when calculating localized Wiener indices. This approach is not utilized by the baseline methods (Dijkstra, Bellman-Ford, Floyd-Warshall), which leads to their suboptimal performance in terms of speed.
>
> In the case of Reddit and ogbn-proteins with parallel Dijkstra, which are smaller datasets with relatively low workloads compared to ogbn-products, the overhead of parallelization is likely to outweigh the benefits, causing parallel Dijkstra to perform slower than the traditional method. However, the advantages of parallelism become more evident in ogbn-products.
>
> The source of each parallel methods are cited as below.
>
> [1] Lund, Ben, and Justin W. Smith. "A Multi-Stage CUDA Kernel for Floyd-Warshall." arXiv preprint arXiv:1001.4108, 2010.
>
> [2] Harish, Pawan, and Petter J. Narayanan. "Accelerating large graph algorithms on the GPU using CUDA." International conference on high-performance computing. Berlin, Heidelberg: Springer Berlin Heidelberg, 2007.

---

> ### Author Response · Authors · 2024-12-02
> **Gentle Reminder**
>
> Dear Reviewer fkMV,
>
> We sincerely thank the reviewer for the thoughtful suggestions and concerns regarding TopGQ. The comments significantly helped enrich our paper, which led to the following experiments of TopGQ as below.
>
> - We enhanced the literature review regarding state-of-the-art GNN quantization works.
> - We added the comparison results of the localized Wiener Index and other centralities in Section 6.6, as experimental support of our node grouping method.
> - We added further clarification of Scale Absorption in Section 5.3.
> - We added comparison results of TopGQ with its QAT settings in Appendix G as Table 14 and 15.
> - We updated the experimental results of baselines in citation datasets regarding fluctuations in Appendix F, as in Table 13.
> - We revised the quantized inference time measurement in Table 5 in the main paper.
> - We compared the state-of-the-art quantization results with TopGQ with citation datasets.
> - We further compare the parallel all-pair shortest path methods with our accelerated localized Wiener Index calculation algorithm.
>
> The experiments above helped us explore an in-depth analysis of TopGQ, and we deeply thank the reviewer for this enhancement.
>
> As the discussion phase will end soon, we wanted to kindly follow up to confirm whether our response has addressed your concerns. Please feel free to let us know if there are any remaining questions or issues, and we would be happy to provide further clarification or discussion.
>
> We thank the reviewer again for dedicating the time and effort to reviewing our work.
>
> Sincerely, the authors of TopGQ.

---

### Official Review · Reviewer_APES · 2024-11-02

**Soundness:** 3
**Presentation:** 3
**Contribution:** 3
**Rating:** 5
**Confidence:** 5

**Summary:**

This paper proposes TopGQ, a post-training quantization (PTQ) framework for GNNs. Unlike existing quantization methods that rely on quantization-aware training (QAT), which involves retraining the model with gradient updates, TopGQ achieves efficient quantization by leveraging the topological information of the graph without requiring any retraining. Specifically, TopGQ groups vertices with similar topology information, including inward degree and localized Wiener index, to share quantization parameters within the group, which can quantize GNNs without backpropagation and accelerate the quantization. To further optimize inference efficiency, TopGQ absorbs group-wise scale factors into the adjacency matrix during aggregation steps, which allows for efficient integer matrix multiplication. Experiments show that TopGQ outperforms SOTA GNN quantization methods in performance with a faster quantization speed.

**Strengths:**

- The proposed method is innovative. Both indegree information and localized Wiener index are used for node grouping to effectively address the high feature magnitude variance issue in GNN quantization.
- Accelerating the quantization process of GNNs is necessary.

**Weaknesses:**

- This work may not achieve a wall-clock speedup on graph-level tasks because it requires grouping the nodes of unseen graphs and computing quantization parameters.
- The application of this method is limited. Because of the use of scale absorption, it seems hard to apply this method to GAT models.
- The acceleration of the Accelerated Wiener Index Computation Algorithm is mainly because of parallel computing rather than the proposed algorithm.
- The third method, scale absorption, is commonly used in network quantization. It should not be a key contribution to this paper.

**Questions:**

- In Table 6, the baseline methods use the SciPy implementation. It is better to compare the Accelerated Wiener Index Computation Algorithm with a parallel Dijkstra method.
- Can this method be applied to GAT models? If so, it is better to add more experiments about GAT quantization to show the generalization of TopGQ.
- In the experiments, the baseline $A^2Q$ method also uses a uniform quantization, but it is a mixed-precision method. So it would be better to compare a mixed-precision version of $A^2Q$ under the same compression or computation constraint.

---

> ### Author Response · Authors · 2024-11-22
>
> ### **W1. This work may not achieve a wall-clock speedup on graph-level tasks because it requires grouping the nodes of unseen graphs and computing quantization parameters.**
>
> | k   | Dataset   | GCN Test Inference (s) | Overhead (s) | Proportion |
> |-----|-----------|:----------------------------------:|:---------------------------------------:|:------------:|
> | **2**   | **Proteins**  | 0.0438                          | 0.0001                                | 0.23%      |
> |     | **NCI1**      | 0.13321                         | 0.0003                                | 0.22%      |
> | **3**   | **Proteins**  | 0.0438                          | 0.0018                                | 4.07%      |
> |     | **NCI1**      | 0.13321                         | 0.0048                                | 3.63%      |
>
>
> $\to$ We provide the inference time for the setting concerned by the reviewer with graph datasets, Proteins and NCI1, for the inductive setting. As we can see in the table, the overhead required to compute the topological information of unseen test nodes accounts for only a very small portion(smaller than 1%) of the total inference time. This is enabled by localized Wiener Index acceleration by TopGQ.
>
> ### **W2. The application of this method is limited. Because of the use of scale absorption, it seems hard to apply this method to GAT models. Q2. Can this method be applied to GAT models? If so, it is better to add more experiments about GAT quantization to show the generalization of TopGQ.**
>
> $\to$ The reason is that the GAT’s characteristic attention-based edge weights require dynamic quantization, which cannot be precomputed. Please note that this restriction applies not only to ours but also to all GNN quantization methods. However, we modify our proposed scale absorption method for GAT, and provide experimental results below. The experimental results show that TopGQ also performs well in GAT architecture.
>
> **Citation graphs**
> |Method|Type|Bit|Cora Acc.(%)|Q.time(s)|Citeseer Acc.(%)|Q.time(s)|PubMed Acc.(%)|Q.time(s)|
> |-|:-:|-|-:|-|-:|-|-:|-|
> |-|-| FP32 |82.10|-|74.10|-|79.42|-|
> |
> |Degree-Quant|QAT|INT8|81.70|18.30s|69.80|41.31s|79.20|61.02s|
> |A2Q|QAT|INT8|77.50|2.44s|69.50|2.55s|72.80|3.11s|
> |SGQuant|QAT|INT8|79.90|5.71s|68.40|8.72s|76.00|9.74s|
> |TopGQ|PTQ|INT8|82.02|0.86s|73.70|1.11s|79.32|1.26s|
> |
> |Degree-Quant|QAT|INT4|80.70|18.78s|23.10|40.91s|74.50|60.89s|
> |A2Q|QAT|INT4|76.80|2.44s|61.80|2.47s|70.50|3.16s|
> |SGQuant|QAT|INT4|74.70|5.55s|66.20|8.67s|72.40|9.77s|
> |TopGQ|PTQ|INT4|80.34|0.92s|66.92|1.20s|78.06|1.25s|
>
> **Graph classification tasks**
> |Method|Type|Bit|Proteins Acc.(%)|Q.time(s)|NCI1 Acc.(%)|Q.time(s)|
> |-|:-:|-|-:|-|-:|-|
> |-|-| FP32 |75.56|-|79.73|-|
> |
> |Degree-Quant|QAT|INT8|72.41|3580.78s|74.50|7988.47s|
> |A2Q|QAT|INT8|72.42|385.62s|72.28|997.62s|
> |SGQuant|QAT|INT8|68.82|267.65s|74.42|753.09s|
> |TopGQ|PTQ|INT8|75.74|4.87s|79.48|9.49s|
> |
> |Degree-Quant|QAT|INT4|71.96|3626.77s|74.01|8078.41s|
> |A2Q|QAT|INT4|70.36|396.62s|66.16|1002.95s|
> |SGQuant|QAT|INT4|59.56|267.66s|58.49|754.85s|
> |TopGQ|PTQ|INT4|69.09|4.71s|69.70|9.90s|
>
> We also specify how Scale Absorption is applied in GAT models.
> GAT requires quantization of edge weights ($A$) during inference due to the need for FP32 operations to obtain edge weights.
> In GAT, Scale Absorption is implemented right before the run-time quantization process by an FP32 element-wise multiplication between A and the precalculated scales.
>
> ### **W3, Q1: The acceleration of the Accelerated Wiener Index Computation Algorithm is mainly because of parallel computing rather than the proposed algorithm. In Table 6, the baseline methods use the SciPy implementation. It is better to compare the Accelerated Wiener Index Computation Algorithm with a parallel Dijkstra method.**
>
> $\to$ The acceleration of our localized Wiener Index calculation comes from the proposed algorithm and does not mainly come from parallel computing. We modified the algorithm from the idea that the distance of an arbitrary node pair is always bounded by hop-count k, which is a perception that baseline methods (Dijkstra, Bellman-ford, Floyd-Warshall) do not utilize. This theoretical bound reduces excessive searches and computations of feasible paths, enabling great speedup.
>
> We are working on the additional experiment to compare our accelerated Wiener index algorithm against parallel versions of other traditional algorithms. We will promptly notify the reviewer with the updated results as soon as they are available.

---

> ### Author Response · Authors · 2024-11-22
>
> ### **Q3. In the experiments, the baseline $A^2Q$ method also uses a uniform quantization, but it is a mixed-precision method. So it would be better to compare a mixed-precision version of $A^2Q$ under the same compression or computation constraint.**
>
> $\to$ As the reviewer mentioned, we fix the bitwidth of $A^2Q$ to make a comparison under the same compression or computation constraint. While we focus on fast and efficient fixed-precision quantization, we believe comparison on mixed-precision setting is out of our scope.
>
>
> We would like to clarify that the reason we set the baseline $A^2Q$ method to use fixed-precision quantization, was not to win $A^2Q$ in a more advantageous setting, but to illustrate best that TopGQ targets and can address quantization issues that current GNN quantization work faces challenges: A fast and effective fixed-precision quantization method for GNNs, and therefore does not intend nor introduce unfair comparison. We selected $A^2Q$ for a baseline method as it represents one of the latest successful works of GNN Quantization in the separate domain of mixed-precision quantization. We want to additionally note that quantization methods rarely perform across both fields as each aims for distinct application scenarios, including differences in bit-width constraints and deployable hardware.

---

> ### Author Response · Authors · 2024-11-23
>
> We apologize for the inconvenience of our response to W4 unintentionally being omitted from the order of the previous comments. We provide it below.
>
> ### **W4. The third method, scale absorption, is commonly used in network quantization. It should not be a key contribution to this paper.**
>
> $\to$  Scale Absorption is a unique method as its purpose is to preserve integer-format aggregation speedups while mitigating quantization challenges in GNN activations. GNNs have quantization difficulties induced by the nature of activations in GNN layers. Activation outliers occur node-wise due to the message-passing mechanism in GNN layers, where repeated aggregation can amplify values, leading to significant outliers. This observation is illustrated in Figure 5. Its application prevents activations from quantizing activations in a poor feature-wise(column-wise) manner, and ensures integer operations in aggregation.
>
> We are deeply interested in understanding the similarities the reviewer perceives between other quantization methods and Scale Absorption. We look forward to discussing further with the reviewer about the topic.

---

> ### Author Response · Authors · 2024-11-26
>
> As our experiment results are ready, we provide our responses to W3 and Q1 as below.
> ### **W3, Q1. The acceleration of the Accelerated Wiener Index Computation Algorithm is mainly because of parallel computing rather than the proposed algorithm. / Q1. In Table 6, the baseline methods use the SciPy implementation. It is better to compare the Accelerated Wiener Index Computation Algorithm with a parallel Dijkstra method.**
>
> $\to$ We update the baselines with parallel algorithms on GPU for all-pair shortest paths and present its speed performance with a new table.
>
> | Datasets                | Reddit  | ogbn-proteins | ogbn-products |
> |-----------------------|---------|---------------|---------------|
> | **Method**                | **Process Time (h)**|||
> | Dijkstra              | 0.16    | 0.11          | 8.52          |
> | Parallel Dijkstra     | 0.18    | 0.13          | 6.99          |
> | Floyd-Warshall        | 0.57     | 0.41          | 35.44         |
> | Parallel Floyd-Warshall | 0.26   | 0.12          | 1.98          |
> | Ours              | 0.0004  | 0.0002        | 0.2855        |
> | Speedup               | 409.67  | 602.30        | 6.93          |
>
>
>
> By the table, it is clear that the acceleration of the localized Wiener Index computation is from its algorithm, rather than parallel computing. In ogbn-products, both the parallel Dijkstra and parallel Floyd-Warshall methods improve the speed of the traditional approaches but remain slower than our Accelerated Localized Wiener Index computation. This is due to the proposed algorithm with its theoretical bounds, where the distance between any arbitrary node pair is limited by a hop-count of k when calculating localized Wiener indices. This approach is not utilized by the baseline methods (Dijkstra, Bellman-Ford, Floyd-Warshall), which leads to their suboptimal performance in terms of speed.
>
> In the case of Reddit and ogbn-proteins with parallel Dijkstra, which are smaller datasets with relatively low workloads compared to ogbn-products, the overhead of parallelization is likely to outweigh the benefits, causing parallel Dijkstra to perform slower than the traditional method. However, the advantages of parallelism become more evident in ogbn-products.
>
> The source of each parallel methods are cited as below.
>
> [1] Lund, Ben, and Justin W. Smith. "A Multi-Stage CUDA Kernel for Floyd-Warshall." arXiv preprint arXiv:1001.4108, 2010.
>
> [2] Harish, Pawan, and Petter J. Narayanan. "Accelerating large graph algorithms on the GPU using CUDA." International conference on high-performance computing. Berlin, Heidelberg: Springer Berlin Heidelberg, 2007.

---

> ### Author Response · Authors · 2024-12-02
> **Gentle Reminder**
>
> Dear Reviewer APES,
>
> We deeply thank the reviewer for the insightful suggestions and concerns for TopGQ. The valuable comments have been greatly helpful in improving our paper.
> The feedback has significantly contributed to enhancing our revision of TopGQ, leading to conducting new experiments with TopGQ as below.
>
> - We added experimental results of localized Wiener Index calculation cost of unseen nodes in Appendix H as Table 16.
> - We provide quantization results of TopGQ on GAT models in Appendix D.
> - We added further clarification of Scale Absorption in Section 5.3.
> - We further compare the parallel all-pair shortest path methods with our accelerated localized Wiener Index calculation algorithm.
>
> The experiments mentioned above have been instrumental in improving the quality of our work, and we sincerely thank the reviewer for this enhancement.
>
> As the discussion phase is approaching its end, we hope to know if our response has addressed the concerns and questions raised by the reviewer. If there are any remaining issues or if our response falls short, we would be glad to discuss them further.
>
> We thank the reviewer again for generously dedicating valuable time to reviewing our work, TopGQ.
>
> Sincerely, the authors of TopGQ.

---

### Official Review · Reviewer_7KRu · 2024-11-03

**Soundness:** 3
**Presentation:** 3
**Contribution:** 2
**Rating:** 5
**Confidence:** 3

**Summary:**

This paper proposed a post-training quantization method for graph neural network.  It uses both the degree information and the topological information (localized Wiener index) to efficiently group the node with similar embedding magnitude together.  The paper also employed a fast algorithm for calculating the local Wiener index to reduce the quantization and inference overhead.  Experiments show that the method well maintain the accuracy of the models.

**Strengths:**

1. This paper uses post-training quantization without backprop, while the related works are mostly quantization-aware training, or requires backpropagating gradients in the quantization procedure.
2. It is novel to consider topology information to group the nodes and performs well.
3. The method has little harm on the accuracy performance.

**Weaknesses:**

1. The overhead of calculating the topological information is faster compared to other baselines, but it seems still a burden compared with the inference time, especially considering most inference is done in batch.
2. It looks like that the inference is not done in a batched manner (correct me if I'm wrong), making the inference time for fp32 baseline extremely long.  It would be better if batched inference results can be shown and compared to have a full understanding of the capabilities of this method.
3. There is no discussion of combining this method with sampling methods, like neighbor sampling or subgraph sampling.  Real world graph training for node tasks mostly requires sampling.  And the inference for neighbor sampling is much faster.  I would like to see how the method performs in these cases and how large the overheads are.

**Questions:**

1. I wonder how the inference is done, because the inference time on ogbn-products is surprisingly long.  Is it done in a batched manner or each single node goes through the network individually?
2. The accuracy of SAGE on ogbn-products Table 1 is abnormally low.  On the ogbn-leaderboard, SAGE on ogbn-products is over 78%.  Why is this happening?
3. Are there any other indexes that may be better than Wiener index, like the spectral information?
4. Can the authors give more information about why quantization for GNN models is important, or in what scenario is it important?  The largest dataset used here (ogbn-products) can be trained and inferred in the full-graph manner in one single card (A6000), which is much faster than the inference time number shown here.

I would like to raise my score if my concerns are properly addressed.

---

> ### Author Response · Authors · 2024-11-22
>
> ### **W1. The overhead of calculating the topological information is faster compared to other baselines, but it seems still a burden compared with the inference time, especially considering most inference is done in batch.**
>
> $\to$ We want to emphasize that computing localized Wiener indices is done only once for each graph during the quantization time. Then, during inference, we only need to calculate the Wiener indices on the unseen nodes. To compare this overhead with the inference time, we provide comparison results in GCN INT8 settings with the graph datasets PROTEINS and NCI1. The nodes from the test set graphs will be the unseen nodes, and their information will have to be calculated during inference time.
> | k   | Dataset   | GCN Test Inference (s) | Overhead (s) | Percentage |
> |-----|-----------|:----------------------------------:|:---------------------------------------:|:------------:|
> | **2**   | **Proteins**  | 0.0438                          | 0.0001                                | 0.23%      |
> |     | **NCI1**      | 0.13321                         | 0.0003                                | 0.22%      |
> | **3**   | **Proteins**  | 0.0438                          | 0.0018                                | 4.07%      |
> |     | **NCI1**      | 0.13321                         | 0.0048                                | 3.63%      |
>
>
>
> As we can see in the table, the overhead for calculating Wiener index of unseen test nodes accounts for only a very small portion(smaller than 1%) of the total inference time. Note that this is made possible from a specialized algorithm to accelerate the computation of the localized Wiener Index, which is another contribution of TopGQ. We added this analysis in Section G.
>
>
>
> ### **W2, W3, Q1: How is inference performed in your method, particularly for the FP32 baseline, where the inference time seems extremely long? Is it conducted in a batched manner or node-by-node? Additionally, have you considered evaluating the method with sampling techniques (e.g., neighbor or subgraph sampling) to improve inference efficiency**
>
>
> $\to$ Our unoptimized kernel has caused the long inference time measure. We improved our inference kernel to resolve speed concerns and, therefore, provide a new table that presents enhanced full-batch inference time with practical durations. This updated table is now included as Table 5 in the revised version of TopGQ.
>
>
> |Method|Type|Bit|Reddit (s)|Speedup|OGBN-Products (s)|Speedup|
> |-|-|-|-:|-:|-:|-:|
> |-|-|FP32|1.41|-|1.45|-|
> |Degree-Quant|QAT|INT8|1.22|1.15$\times$|1.30|1.12$\times$|
> |A2Q|QAT|INT8|1.30|1.08$\times$|1.78|0.82$\times$|
> |SGQuant|QAT|INT8|1.25|1.13$\times$|1.31|1.11$\times$|
> |TopGQ|PTQ|INT8|1.24|1.13$\times$|1.30|1.11$\times$|
>
> We can see that along the baselines, TopGQ can provide speedups from integer operations.
> We can also use sampling methods with TopGQ batched inference. We measured the batched inference with neighbor sampling with a size factor of [25, 10], and mini-batch size 4096. We compared its inference time with the full-batch inference, and the baselines likewise.
>
>
> |Dataset|Method|Inference Time|Slowdown compared to full-batch|
> |-|-|-:|-:|
> |**Reddit**|Degree-Quant QAT|1.47s|0.83$\times$|
> ||A2Q QAT|2.23s|0.59$\times$|
> ||SGQuant QAT|1.52s|0.82$\times$|
> ||TopGQ PTQ|1.49s|0.83$\times$|
> |**ogbn-products**|Degree-Quant QAT|35.36s|0.037$\times$|
> ||A2Q QAT|54.46s|0.033$\times$|
> ||SGQuant QAT|36.30s|0.036$\times$|
> ||TopGQ PTQ|35.77s|0.036$\times$|
>
> We can observe that the batched inference time of TopGQ with sampling methods has a consistent amount of slowdown, compared to other methods except $A^2Q$, whose inference speed differs as it processes quantization parameter search in run-time. This guarantees that the overhead of applying sampling methods to TopGQ inference remains consistent with other GNN quantization methods.
>
> ### **Q2. The accuracy of SAGE on ogbn-products Table 1 is abnormally low. On the ogbn-leaderboard, SAGE on ogbn-products is over 78%. Why is this happening?**
>
> $\to$ We believe the accuracy difference comes from the choice of aggregator functions. In the leaderboard, the selected aggregator function was “mean”, while our setting selected “max” as the aggregator function in the experiments.
>
> We would like to present the quantization results of TopGQ for graphSAGE architecture with “mean” aggregators, with FP32 accuracies comparable to those of ogbn-leaderboard scores. As presented in the table, TopGQ can preserve performance regardless of aggregator functions. We added these results in Section E of the revision.
>
> |Method|Bit| Acc. (%)|Q.time (s)|
> |-|:-:|-:|-:|
> |-| FP32 |79.00|-|
> |
> |Degree-Quant|INT8|78.73|482702.99|
> |A2Q|INT8|77.43|56232.71|
> |SGQuant|INT8|41.22|114816.30|
> |TopGQ|INT8|77.17|436.23|
> |
> |Degree-Quant|INT4|75.65|493003.42|
> |A2Q|INT4|45.88|58034.59|
> |SGQuant|INT4|26.95|118242.50|
> |TopGQ|INT4|70.23|435.12|

---

> ### Author Response · Authors · 2024-11-22
>
> ### **Q3. Are there any other indexes that may be better than Wiener index, like the spectral information?**
>
> $\to$ To further identify the advantages that localized Wiener Index has for quantization, we compared the quantization results with proteins and nci1 datasets over other graph properties such as betweenness centrality, closeness centrality, and Katz centrality. The results are as below.
>
> The other node centrality measures depict suboptimal performance compared to using localized Wiener Index, in both INT4 and INT8 settings.
> We believe that the result stems from the unique expressiveness of the localized Wiener Index in capturing local compactness of a node within k-hop neighbors: A small value of a node indicates a dense connectivity within its neighbors, and relatively rapid propagation of features via message passing. Therefore, TopGQ can effectively group node features with distinctive ranges, as shown in Figure 2 in the paper, leading to enhanced quantization quality.
>
> |Method|Bit|Proteins GCN|Proteins GIN|Proteins GS|NCI1 GCN|NCI1 GIN|NCI1 GS|
> |-|-|-:|-:|-:|-:|-:|-:|
> |-| FP32 |76.19|74.79|72.87|80.41|81.46|78.46|
> |
> |Degree Centrality only|INT8|72.57|71.86|70.48|78.91|81.28|78.32|
> |+ Betweenness Centrality|INT8|62.10|61.55|55.08|76.89|75.18|75.13|
> |+ Closeness Centrality|INT8|62.48|64.96|57.33|76.49|76.68|75.85|
> |+ Katz Centrality|INT8|56.82|57.97|48.56|64.20|62.19|64.27|
> |+ Ours|INT8|75.94|74.86|74.00|80.91|81.88|79.16|
> |
> |Degree Centrality only|INT4|56.15|45.04|50.65|60.54|69.71|75.46|
> |+ Betweenness Centrality|INT4|59.03|54.25|50.58|63.81|67.55|70.61|
> |+ Closeness Centrality|INT4|58.52|61.73|50.48|63.14|69.54|71.97|
> |+ Katz Centrality|INT4|53.68|55.24|44.08|57.19|57.36|57.77|
> |+ Ours|INT4|70.15|70.61|69.67|67.53|78.49|76.43|
>
> We experimented with other node centralities to substitute the localized Wiener Index and provide quantization results based on new grouping methods with PROTEINS and NCI1 datasets.
> Results show that the localized Wiener Index performs significantly better than other information centralities, especially in low-bit settings.
>
> ### **Q4. Can the authors give more information about why quantization for GNN models is important, or in what scenario is it important? The largest dataset used here (ogbn-products) can be trained and inferred in the full-graph manner in one single card (A6000), which is much faster than the inference time number shown here.**
>
> $\to$ GNN quantization addresses the high memory and computational demands of large-scale graph processing, a unique challenge in GNNs compared to other neural networks, as hardware requirements for GNN model inferences are significantly more sensitive to data size. It enables broader deployment of GNNs, especially in domains like traffic forecasting [1], IoT [2], bioinformatics [3], and knowledge graphs [4], where scalability to their real-world large graphs is critical. By compressing GNN models with minimal performance loss, quantization broadens the usage and deployment of GNNs in resource-constrained systems.
>
> We want to add that exploration of improving model performance with large-scale graphs is in its infancy in the GNN quantization field. we reported the performance on the ogbn-product dataset (a graph with over 2,400,000 nodes), which was a scale not experimented on baseline works of GNN quantization. We believe expanding quantization to GNNs trained with larger-scale graphs than ogbn-products is a future work to continue, and will contribute to extensive usage of GNN models in real-world.
>
> [1] Jiang, Weiwei, and Jiayun Luo. "Graph neural network for traffic forecasting: A survey." Expert systems with applications 207 (2022): 117921.
>
> [2] Dong, Guimin, et al. "Graph neural networks in IoT: A survey." ACM Transactions on Sensor Networks 19.2 (2023): 1-50.
>
> [3] Li, Yu, et al. "Deep learning in bioinformatics: Introduction, application, and perspective in the big data era." Methods 166 (2019): 4-21.
>
> [4] Chen, Huiyuan, et al. "Tinykg: Memory-efficient training framework for knowledge graph neural recommender systems." Proceedings of the 16th ACM Conference on Recommender Systems. 2022.
>
> [5] Rahmani, Saeed, et al. "Graph neural networks for intelligent transportation systems: A survey." IEEE Transactions on Intelligent Transportation Systems 24.8 (2023): 8846-8885.

---

> ### Author Response · Authors · 2024-12-02
> **Gentle Reminder**
>
> Dear Reviewer 7KRu,
>
> We sincerely appreciate the reviewer’s thoughtful concerns and suggestions regarding TopGQ, which we believe have provided valuable guidance for further enhancing our paper.
> The comments helped enrich our revised version of TopGQ, leading to the following modifications in our paper.
> - We added experimental results of localized Wiener Index calculation cost of unseen nodes in Appendix H as Table 16.
> - We revised the quantized inference time measurement in Table 5 in the main paper.
> - We provide quantization results of TopGQ on GraphSAGE with “mean” aggregators in Appendix E.
> - We added the comparison results of the localized Wiener Index and other centralities in Section 6.6, as experimental support of our node grouping method.
>
> The above experiments have greatly contributed to enhancing the quality of our work, and we deeply thank the reviewer for this improvement.
>
> As the discussion phase is ending soon, we would appreciate it if the reviewer could let us know whether our response has addressed the concerns raised regarding the paper. We would gladly answer any additional questions if the response is insufficient, and the reviewer has unresolved concerns.
>
> We thank the reviewer again for dedicating the time and effort to reviewing our work.
>
> Sincerely, the authors of TopGQ.

---

### Official Review · Reviewer_PWvf · 2024-11-08

**Soundness:** 1
**Presentation:** 1
**Contribution:** 2
**Rating:** 3
**Confidence:** 4

**Summary:**

The paper presents a post-training quantization (PTQ) framework tailored for graph neural networks (GNNs) that mitigates the quantization error by grouping nodes based on topology (indegree and local Wiener index) and absorbing group-specific scales into the adjacency matrix, TopGQ enables highly efficient integer matrix multiplication. Experiments demonstrate that TopGQ achieves faster quantization speeds compared to quantization-aware training (QAT) methods.

**Strengths:**

1-TopGQ introduces a PTQ approach, eliminating the need for gradient computation, which reduces quantization time significantly.

2-Using indegree and local Wiener index to group nodes based on topological similarity seems novel

**Weaknesses:**

1- Lack of motivation and not providing proper application for the work

2-Considerable accuracy drop in 4-bit scenarios.

3-Less detailed justification about the obtained results.

**Questions:**

1- The motivation of the paper is not clear to me. The paper needs to provide applications where fast quantization is urgently needed. From the plots, the QAT takes about 2 hours for large workloads in most cases. Since it needs to be done one time, quantization time cannot be a good motivation to me especially when the proposed method has an accuracy drop. Please provide more applications and cite a few notable references that show quantization time matters. It would be helpful to suggest specific applications or scenarios where fast quantization could be particularly valuable.

2- The motivation behind quantization can be time and storage. However, as Table 5 shows, TOPGQ is not showing any gains in terms of inference time. No results for storage reduction are provided as well. The authors can provide a more comprehensive analysis of the trade-offs between accuracy, quantization time, inference time, and storage requirements.

3- The author didn’t study GAT. Is there any reason behind it?

4- Several studies show that PTQ starts degrading performance when it goes below 4 bits. I think the proposed PTQ technique will not work well compared to QAT in below 4-bit unless the authors show some results that nullify the hypothesis. Even for 4-bit, I can see quite a notable accuracy drop compared to fp32 (e.g., around 7% Reddit GCN and 21% on ogbnproducts GCN). The authors need to provide an application where such drops are acceptable.

5-The author needs to provide detailed justification where PTQ can perform better than QAT and FP32. For example, why in the case of ogbnproteins, TopGQ is better than FP32 by a noticeable margin?

6-How sensitive is TopGQ to changes in group sizes or to variations in the rank used in low-rank approximations?

7- Outliers might still exist within topologically grouped nodes, especially in large-scale graphs. How does this affect quantization quality?

8- Table 5 needs to provide results for 4-bit as well.

9- An ablation study comparing the use of indegree and Wiener index with other centrality measures (e.g., closeness or betweenness) would provide insights into the robustness of TopGQ’s topology-based grouping.

10- An analysis of how changing grouping parameters (e.g., group size, hop count for Wiener index) affects quantization error would clarify the stability and adaptability of TopGQ.

11-The provided code is just .py file without any instructions on how to run and get results. That limits the reproducibility.
The could should provide a README file with setup instructions, example commands, and a requirements.txt file for dependencies.

12-Minor Typos:

A– Consider changing "huge" to " significant, substantial, etc"

B- Change "On the quantization time" to "In terms of quantization time."

C- "a fair comparison, we use fixed-precision quantization" – Add "a" before "fixed-precision quantization"

D. with a significant drop in accuracy of 9.0%p!

---

> ### Author Response · Authors · 2024-11-22
>
> ### **W1. Lack of motivation and not providing proper application for the work.**
>
> We would like to address the issue of weak motivation and application studies of fast GNN quantization by answering related questions (Q1, Q2) as below.
>
> **Q1. The paper needs to provide applications where fast quantization is urgently needed. Please provide more applications and cite a few notable references that show quantization time matters.**
>
> $\to$ We provide some typical applications that need fast quantization:
>
> - GNNs processing temporal graphs with rapid modification over time [1,2]
> - GNNs in edge-device-enabled transportation systems [3,4], and recommendation systems [5,6]
> - GNNs in anomaly-sensitive program designs such as fraud detection in financial transactions [7,8,9]
>
> Despite the state-of-the-art performance of GNNs, the field still has limited usage in practical applications due to the growing size of real-world graphs. Especially with user-end devices with limited computational power and memory, GNNs have to be tailored or compressed to accommodate the hardware constraints. However, real-world graphs such as traffic network graphs or social media graphs are continuously updated, which requires multiple compressing of the GNNs to keep the application up-to-date. In such cases, fast quantization of GNN can be the only viable solution.
> The issue can be more severe in safety-critical applications such as edge fraud detection in financial transactions, where among the discovery of a new security flaw, a rapid update for safety patch is required. We believe GNNs in these domains are especially sensitive to changes in both the graph and the model and thus require fast adaptations, extremely to a real-time level.
>
> We added this in the revised version of our paper, in the introduction section.
>
> [1] Gao, Shihong, et al. "ETC: Efficient Training of Temporal Graph Neural Networks over Large-scale Dynamic Graphs." Proceedings of the VLDB Endowment 17.5 (2024): 1060-1072.
>
> [2] Longa, A., et al. "Graph Neural Networks for temporal graphs: State of the art, open challenges, and opportunities." TRANSACTIONS ON MACHINE LEARNING RESEARCH (2023).
>
> [3] Jiang, Weiwei, and Jiayun Luo. "Graph neural network for traffic forecasting: A survey." Expert systems with applications 207 (2022): 117921.
>
> [4] Sharma, Amit, et al. "A graph neural network (GNN)-based approach for real-time estimation of traffic speed in sustainable smart cities." Sustainability 15.15 (2023): 11893.
>
> [5] Gao, Chen, et al. "A survey of graph neural networks for recommender systems: Challenges, methods, and directions." ACM Transactions on Recommender Systems 1.1 (2023): 1-51.
>
> [6] Yao, Yuhang, et al. "FedRule: Federated rule recommendation system with graph neural networks." Proceedings of the 8th ACM/IEEE Conference on Internet of Things Design and Implementation. 2023.
>
> [7] Lu, Mingxuan, et al. "Bright-graph neural networks in real-time fraud detection." Proceedings of the 31st ACM International Conference on Information & Knowledge Management. 2022.
>
> [8] Zhou, Hongkuan, et al. "Accelerating large scale real-time GNN inference using channel pruning." arXiv preprint arXiv:2105.04528 (2021).
>
> [9] Liu, Ziqi, et al. "Heterogeneous graph neural networks for malicious account detection." Proceedings of the 27th ACM international conference on information and knowledge management. 2018.

---

> ### Author Response · Authors · 2024-11-22
>
> **Q2. The motivation behind quantization can be time and storage. The authors can comprehensively analyze the trade-offs between accuracy, quantization time, inference time, and storage requirements.**
>
>
> |**Metrics**|**Accuracy**|**Inference Time(s)**|**Inference Speedup**|**Theoretical Cost**|**Quant. Time(h)**|**Quant. Speedup**|**Theoretical Storage**|
> |-|-|-|-|-|-|-|-|
> |A(FP32)|78.41%|1.450|1|$O_{FP}(N^2F_1+NF_1F_2)$|-|-|$O_{FP}(E+F_1F_2+NF_0)$|
> |B(DQ)|75.26%|1.295|1.120|$O_{INT}(N^2F_1+NF_1F_2)+O_{FP_{elem}}(NF_2)$|95.95|1|$O_{INT}(E+F_1F_2+NF_0)+O_{FP}(1)$|
> |B(DQ-PTQ)|46.57%|1.294|1.121|$O_{INT}(N^2F_1+NF_1F_2)+O_{FP_{elem}}(NF_2)$|0.28|343|$O_{INT}(E+F_1F_2+NF_0)+O_{FP}(1)$|
> |C(TopGQ)|76.94%|1.304|1.112|$O_{INT}(N^2F_1+NF_1F_2)+O_{FP_{elem}}(NF_2)$|0.34|282|$O_{INT}(E+F_1F_2+NF_0)+O_{FP}(N_T+F_2)$|
>
>
> - $O_{FP}()$: complexity for floating-point operations / Storage complexity for floating-point values
> - $O_{FP_{elem}}()$: complexity for elementwise floating-point operations
> - $O_{INT}()$: complexity for fixed-point operations / Storage complexity for fixed-point values
>
> As shown in the table, TopGQ finds a good balance between reducing quantization time and preserving accuracy, while other choices in (A), (B), (C) demonstrate disadvantages in either accuracy, time, or memory. (A) suffers from the expensive costs of computation and storage. While (B) alleviates this cost via quantization, the long quantization time is required to obtain the benefits. (C) is free from the quantization time problem but at the cost of huge performance degradation. TopGQ aims to find the best way of addressing each issue by leveraging topological node similarities with an additional amount of storage cost.
>
>
> As for the theoretical costs, we assume GNN layer propagation as AXW operation, with $A \in \mathbb{R}^{N*N}, X \in \mathbb{R}^{N*F1}, W \in \mathbb{R}^{F1*F2}$ with initial dataset size of N * F0. The computation cost shows that quantization converts the expensive floating-point matrix multiplication into integer operations. The additional floating-point cost of (B)~(D) comes from converting integer outputs back to floating-point values.
>
> The measurement is provided by an improved version of our kernel, and the theoretical analysis is based on [1]. We added this in Section C of the revision.
>
> [1] Zhu, Zeyu, et al. "$\rm A^ 2Q $: Aggregation-Aware Quantization for Graph Neural Networks." arXiv preprint arXiv:2302.00193 (2023).
>
> ### **W2, W3 &Q4. Considerable accuracy drop in 4-bit scenarios and less detailed justification about the obtained results: I think the proposed PTQ technique will not work well compared to QAT in below 4-bit unless the authors show some results that nullify the hypothesis. The authors need to provide an application where such drops are acceptable.**
>
> TopGQ can be effective in cases where fast inference speed of GNNs is much more prioritized, such as real-time applications in:
> - point cloud processing based tasks such as indoor navigation, shape modeling, and 3D object detection. [1,2]
> - analyzing high energy physics in particle physics, where GNNs decide whether to collect or discard data from a particle collider within nanoseconds to capture vital information [3]
> - ride-hailing platforms that have to process real-time surrounding traffic data and physical environments for event prediction [4]
>
>
> In low-bit quantization, accuracy drop to some degree is inevitable when compared to FP32 because it essentially limits model capacity. While INT4 TopGQ shows degradation to FP32 in some cases, it shows a clear advantage over baselines in both accuracy and quantization time, suggesting a better option than existing methods. Finally, we emphasize that reaching the performance of FP32 is a shared goal for all quantization methods, and we aim to further close this gap in low-bit settings in our future work.
>
>
> [1] Shao, Jiawei, et al. "Branchy-GNN: A device-edge co-inference framework for efficient point cloud processing." ICASSP 2021-2021 IEEE International Conference on Acoustics, Speech and Signal Processing (ICASSP). IEEE, 2021.
>
> [2] Shi, Weijing, and Raj Rajkumar. "Point-gnn: Graph neural network for 3d object detection in a point cloud." Proceedings of the IEEE/CVF conference on computer vision and pattern recognition. 2020.
>
> [3] Iiyama, Yutaro, et al. "Distance-weighted graph neural networks on FPGAs for real-time particle reconstruction in high energy physics." Frontiers in big Data 3 (2021): 598927.
>
> [4] Luo, Wenjuan, et al. "Dynamic heterogeneous graph neural network for real-time event prediction." Proceedings of the 26th ACM SIGKDD international conference on knowledge discovery & data mining. 2020.

---

> ### Author Response · Authors · 2024-11-22
>
> ### **Q3. The author didn’t study GAT. Is there any reason behind it?**
>
> $\to$ The reason is that the GAT’s attention-based edge weights are computed at runtime, therefore quantization scales of the adjacency matrix are also computed at runtime, meaning our method of absorbing the scale to the adjacency matrix cannot be precomputed. However, scale absorption can be modified to accommodate such dynamic quantization scenarios, which we provide in the below table. The results show that TopGQ also performs well in GAT architecture.
>
> **Citation graphs**
> |Model|Method|Prec.|Cora Acc.(%)|Q.time(s)|Citeseer Acc.(%)|Q.time(s)|PubMed Acc.(%)|Q.time(s)|
> |-|:-:|-|-:|-|-:|-|-:|-|
> |FP32|-|-|82.10|-|74.10|-|79.42|-|
> |
> |Degree-Quant|QAT|INT8|81.70|18.30s|69.80|41.31s|79.20|61.02s|
> |A2Q|QAT|INT8|77.50|2.44s|69.50|2.55s|72.80|3.11s|
> |SGQuant|QAT|INT8|79.90|5.71s|68.40|8.72s|76.00|9.74s|
> |TopGQ|PTQ|INT8|82.02|0.86s|73.70|1.11s|79.32|1.26s|
> |
> |Degree-Quant|QAT|INT4|80.70|18.78s|23.10|40.91s|74.50|60.89s|
> |A2Q|QAT|INT4|76.80|2.44s|61.80|2.47s|70.50|3.16s|
> |SGQuant|QAT|INT4|74.70|5.55s|66.20|8.67s|72.40|9.77s|
> |TopGQ|PTQ|INT4|80.34|0.92s|66.92|1.20s|78.06|1.25s|
>
> **Graph classification tasks**
> |Model|Method|Prec.|Proteins Acc.(%)|Q.time(s)|NCI1 Acc.(%)|Q.time(s)|
> |-|:-:|-|-:|-|-:|-|
> |FP32|-|-|75.56|-|79.73|-|
> |
> |Degree-Quant|QAT|INT8|72.41|3580.78s|74.50|7988.47s|
> |A2Q|QAT|INT8|72.42|385.62s|72.28|997.62s|
> |SGQuant|QAT|INT8|68.82|267.65s|74.42|753.09s|
> |TopGQ|PTQ|INT8|75.74|4.87s|79.48|9.49s|
> |
> |Degree-Quant|QAT|INT4|71.96|3626.77s|74.01|8078.41s|
> |A2Q|QAT|INT4|70.36|396.62s|66.16|1002.95s|
> |SGQuant|QAT|INT4|59.56|267.66s|58.49|754.85s|
> |TopGQ|PTQ|INT4|69.09|4.71s|69.70|9.90s|
>
> In our modified scale absorption for GAT, the absorption is performed at runtime right before the quantization operation, simply by adding an FP32 element-wise multiplication between the adjacency matrix and the precalculated scales. We clarify this in Section D in the revised paper.
>
> ### **Q5. The author needs to provide detailed justification where PTQ can perform better than QAT and FP32. For example, why in the case of ogbn-proteins, TopGQ is better than FP32 by a noticeable margin?**
>
> $\to$ The reason TopGQ can outperform QAT baselines is that our proposed topology-aware node grouping helps to find better quantization parameters. While QAT has an upper hand in the fact that they can train the weights, the existing QAT baselines do not consider the nature of GNN and minimize the quantization error of each node feature. On the other hand, our method directly integrates the nature of GNN aggregation into the quantization parameters by grouping nodes by their k-hop topological structure. Therefore, we believe the superior performance of TopGQ is due to a better ability to find quantization parameters, and is orthogonal to the PTQ/QAT differences.
> In other words, while TopGQ is implemented for PTQ for efficiency, it can bring performance gain in both scenarios. To validate this, we present two variants: Degree-Quant-PTQ and TopGQ-QAT, which are the PTQ and QAT versions of each method, respectively. The experimental results are shown in the table below. The results show that our proposed topology-aware grouping shows better performance regardless of PTQ and QAT.
>
> |INT4|Cora GCN|Cora GIN|Cora GS|Pubmed GCN|Pubmed GIN|Pubmed GS|
> |-|-:|-:|-:|-:|-:|-:|
> |Degree-Quant|79.00|71.90|73.50|78.60|76.60|78.20|
> |Degree-Quant-PTQ|78.42|30.46|78.54|78.34|50.20|77.64|
> |TopGQ-QAT|80.08|76.30|76.64|78.50|77.00|76.72|
> |Original TopGQ|81.50|78.58|79.64|79.58|77.70|79.00|
>
> As for the results that outperform FP32 accuracies, we believe this phenomenon often occurs when the low-bit format is sufficient to handle the original model complexity. We cite some papers that exhibit the mentioned occasion in their experiments. For example, some of [1] and [2] report better performance in 8-bit settings than FP32 settings at various tasks.
>
> We added this explanation in Section 6.2 and Appendix I in the revision.
>
> [1] Wu, Di, et al. "Easyquant: Post-training quantization via scale optimization." arXiv preprint arXiv:2006.16669. 2020.
> [2] Shomron, Gil, et al. "Post-training sparsity-aware quantization." NeurIPS. 2021.

---

> ### Author Response · Authors · 2024-11-22
>
> ### **Q6 & Q10. How sensitive is TopGQ to changes in group sizes or to variations in the rank used in low-rank approximations? An analysis of how changing grouping parameters (e.g., group size, hop count for Wiener index) affects quantization error would clarify the stability and adaptability of TopGQ.**
>
> $\to$ We conducted sensitivity studies regarding the value of hop count k for the Wiener Index, and the results of graph datasets are included in the original paper as Table 7. Additionally, we prepared the study with citation datasets for generalization. The tables are as below.
>
> |k|Prec.|Cora|GCN|GIN|GS|Citeseer|GCN|GIN|GS|PubMed|GCN|GIN|GS|
> |-|-|-|-:|-:|-:|-|-:|-:|-:|-|-:|-:|-:|
> |1|INT4||81.00|77.12|78.82||71.86|69.10|70.86||78.20|75.42|78.50|
> |2|INT4||81.50|78.58|79.64||71.90|70.14|71.76||79.58|77.70|79.00|
> |3|INT4||81.56|78.32|78.56||72.12|70.47|71.38||79.20|77.00|78.72|
> |
> |1|INT8||81.96|78.36|79.92||72.24|70.18|71.84||80.18|78.34|78.90|
> |2|INT8||82.08|78.42|80.30||72.28|70.26|71.96||80.30|78.62|78.94|
> |3|INT8||82.10|78.38|79.54||72.24|70.60|71.92||80.24|78.68|78.84|
>
>
> |k|Prec.|Proteins|GCN|GIN|GS|NCI1|GCN|GIN|GS|
> |-|-|-|-:|-:|-:|-|-:|-:|-:|
> |1|INT4||60.86|51.04|65.77||62.68|70.91|75.90|
> |2|INT4||66.06|63.96|67.01||66.14|77.33|76.50|
> |3|INT4||70.15|70.61|69.67||65.09|78.49|76.43|
> |
> |1|INT8||73.34|72.88|73.03||80.81|81.60|78.88|
> |2|INT8||76.05|74.61|74.22||80.86|81.84|79.10|
> |3|INT8||75.94|74.86|74.00||80.91|81.88|79.16|
>
> In the table, we can observe that TopGQ shows stability in overall quantization performance, with a preference of value k for better accuracy results for several settings.
>
> As for the question of low-rank approximations, TopGQ does not have techniques associated with the concept. Can we kindly ask the reviewer to give more details about the question? Scale Absorption may have brought confusion as it depicts a similar illustration, therefore we include additional clarification of its method below.
>
> Scale Absorption enables node-wise quantization of $X$ in the matrix multiplication $A \times X$ to preserve the precision of $X$ during quantization. Repetitive aggregation in GNN layers induces node-wise outlier activations, as seen in Figures 5 and 6 of our paper. Feature-wise (column-wise) quantization of $X$, required for integer matrix multiplication, degrades precision due to outlier-influenced scales in each column. To address this, Scale Absorption integrates the precalculated quantization scale of $X$ into the edge weights of $A$ before inference, allowing efficient multiplication between quantized $A$ and node-wise quantized $X$.
>
>
> ### **Q7. Outliers might still exist within topologically grouped nodes, especially in large-scale graphs. How does this affect quantization quality?**
>
> $\to$
> We are currently working on illustrating the connection between outliers and quantization quality in large-graph datasets, as we strongly believe the context will provide deeper insights about the performance of TopGQ and enhance its understanding.
> We will promptly let the reviewer know as soon as the analysis is ready to report.
>
> ### **Q8. Table 5 needs to provide results for 4-bit as well.**
>
> $\to$ Practical INT4 deployment is currently a challenge due to the lack of public INT4 sparse matrix multiplication (SPMM) kernels supporting channels-wise asymmetric quantization. Note that building a kernel that can fully utilize the INT4 support of TensorCore (which does not support SPMM operation) is difficult enough to be recognized as a contribution worth a paper. [1] has succeeded in building an INT4 kernel, but only supports naive per-tensor symmetric quantization, and needs major modification to accommodate ours.
>
> Nevertheless, Table 5 of TopGQ shows that TopGQ can accelerate inference in integer formats when paired with appropriate kernels supporting integer operations. Additionally, we improved our inference kernel to resolve speed concerns, and therefore provide a new table that presents enhanced full-batch inference time. This updated table is now included as Table 5 in the revised version of TopGQ.
>
> |Method|Type|Bit|Reddit (s)|Speedup|OGBN-Products (s)|Speedup|
> |-|-|-|-:|-:|-:|-:|
> ||-|FP32|1.41|-|1.45|-|
> |Degree-Quant|QAT|INT8|1.22|1.15$\times$|1.30|1.12$\times$|
> |A2Q|QAT|INT8|1.30|1.08$\times$|1.78|0.82$\times$|
> |SGQuant|QAT|INT8|1.25|1.13$\times$|1.31|1.11$\times$|
> |TopGQ|PTQ|INT8|1.24|1.13$\times$|1.30|1.11$\times$|
>
>
> [1] Wang, Yuke, Boyuan Feng, and Yufei Ding. "QGTC: accelerating quantized graph neural networks via GPU tensor core." Proceedings of the 27th ACM SIGPLAN symposium on principles and practice of parallel programming. 2022.
>
> We anticipate that future kernels leveraging INT4 operations will better support efficient GNN inference.

---

> ### Author Response · Authors · 2024-11-22
>
> ### **Q9. An ablation study comparing the use of indegree and Wiener index with other centrality measures (e.g., closeness or betweenness) would provide insights into the robustness of TopGQ’s topology-based grouping.**
>
> $\to$ To further identify the advantages that localized Wiener Index has for quantization, we compared the quantization results with PROTEINS and NCI1 datasets over other graph properties such as betweenness centrality, closeness centrality, and Katz centrality. The results are as below.
>
> The other node centrality measures depict suboptimal performance compared to using localized Wiener Index, in both INT4 and INT8 settings.
> We believe that the result stems from the unique expressiveness of the localized Wiener Index in capturing local compactness of a node within k-hop neighbors: A small value of a node indicates a dense connectivity within its neighbors, and relatively rapid propagation of features via message passing. Therefore, TopGQ can effectively group node features with distinctive ranges, as shown in Figure 2 in the paper, leading to enhanced quantization quality. We added this in Section 6.6 in the revised version.
>
> |Method|Bit|Proteins GCN|Proteins GIN|Proteins GS|NCI1 GCN|NCI1 GIN|NCI1 GS|
> |-|-|-:|-:|-:|-:|-:|-:|
> |-|FP32|76.19|74.79|72.87|80.41|81.46|78.46|
> |
> |Degree Centrality only|INT8|72.57|71.86|70.48|78.91|81.28|78.32|
> |+ Betweeness Centrality|INT8|62.10|61.55|55.08|76.89|75.18|75.13|
> |+ Closeness Centrality|INT8|62.48|64.96|57.33|76.49|76.68|75.85|
> |+ Katz Centrality|INT8|56.82|57.97|48.56|64.20|62.19|64.27|
> |+ Ours|INT8|75.94|74.86|74.00|80.91|81.88|79.16|
> |
> |Degree Centrality only|INT4|56.15|45.04|50.65|60.54|69.71|75.46|
> |+ Betweeness Centrality|INT4|59.03|54.25|50.58|63.81|67.55|70.61|
> |+ Closeness Centrality|INT4|58.52|61.73|50.48|63.14|69.54|71.97|
> |+ Katz Centrality|INT4|53.68|55.24|44.08|57.19|57.36|57.77|
> |+ Ours|INT4|70.15|70.61|69.67|67.53|78.49|76.43|
>
>
>
>
> ### **Q11. The provided code is just .py file without any instructions on how to run and get results. That limits the reproducibility. The could should provide a README file with setup instructions, example commands, and a requirements.txt file for dependencies.**
>
> $\to$ We thank the reviewer for the helpful feedback. We have updated the code files with detailed instructions for better reproducibility.
>
> ### **Q12. Minor Typos.**
>
> $\to$ We sincerely thank the reviewer for identifying and informing us of the misprints throughout the paper. We have corrected the typos in the revised version.

---

> ### Author Response · Authors · 2024-11-26
>
> As our experiment results are ready, we provide our responses to Q7 as below.
> ### **Q7. Outliers might still exist within topologically grouped nodes, especially in large-scale graphs. How does this affect quantization quality?**
>
> $\to$ To understand the impact of outliers on the quantization quality of TopGQ, we evaluated how outliers influence GNN layer activation quantization methods across different levels of granularity. In the table, the percentage “k%” represents that features of top k% outlier nodes were excluded from quantization, retaining their original precision as FP32 values.
>
> The experiment was set on INT4 quantization setting with GCN architecture, with dataset Reddit.
>
> | **Method**                          | **Bits** |  |||     |
> |:---------------------------------|:------:|--------:|--------:|--------:|--------:|
> |**Percentage of FP32 Outliers**| | **0%**     | **1%**     | **5%**     | **10%**    |
> | FP32                            |      | 91.91% | 91.91% | 91.91% | 91.91% |
> | No Node Grouping                | INT4 | 6.37%  | 39.36% | 62.50% | 65.05% |
> | Node Grouping with Only Indegree|   INT4   | 78.87% | 80.28% | 81.25% | 82.25% |
> | TopGQ                           |   INT4   | 83.02% | 83.09% | 83.49% | 83.98% |
>
> As shown in the table, quantization without any node grouping strategies experiences significant degradation, with performance improving sharply—up to a 58.68% increase—when more outlier nodes are excluded from quantization. Similarly, quantization with only node indegree information demonstrates a comparable trend, with a smaller accuracy gap of 3.38%. Both settings show relatively high sensitivity to outlier quantization.
>
> In contrast, TopGQ’s node grouping approach exhibits robustness, with an accuracy gap of no more than 1%. This result clearly demonstrates that TopGQ effectively mitigates the impact of outliers on quantization by its node grouping, ensuring stable and high-quality activation quantization even in the presence of extreme values. TopGQ effectively separates and quantizes outliers, maintaining overall quantization quality even with their inclusion in quantization.

---

> > ### Comment · Reviewer_PWvf · 2024-11-27
> >
> > I thank the authors for putting in the effort and providing more results.
> > I think the paper needs major modifications from the initial submission. The authors need to apply the given comments to make the paper stronger. With that, I keep my score.

---

> ### Author Response · Authors · 2024-12-02
>
> Dear Reviewer PWvf,
>
> We would like to thank the reviewer for all the constructive feedback which helped improve and analyze TopGQ in various aspects. We hope that the provided experimental results have addressed the questions and concerns raised by the reviewer regarding TopGQ. We also welcome any additional suggestions the reviewer may have for further modification of our paper.
>
> Thank you again for your time and effort.

---

### Public Comment · ~Samir_Moustafa1 · 2024-11-26
**Runtime Kernel Code for Reproducibility**

Dear Authors,

Thank you for your work on TopGQ. Could you share the code or details of the kernel used to measure runtimes for TopGQ, A²Q, SGQuant, and DQ? This would help verify the benchmarks, especially for A²Q, which reports a high speedup factors within the original paper.

---

> ### Author Response · Authors · 2024-12-02
>
> Dear Samir Moustafa,
>
> We deeply appreciate your interest in our work, TopGQ.
> We would gladly open our kernel source when our paper is accepted.
>
> Sincerely, the authors of TopGQ.

---

### Meta-Review · Area_Chair_tbpK · 2024-12-22

**Metareview:**

This paper proposes a new method for quantizing GNNs without retraining through node grouping. Compared with existing methods like Degree-Quant, A2Q and SGQuant, it has a faster quantization speed at lower cost of accuracy. While the authors have provided extensive experiments to address all the reviewers' concerns during the discussion period, some issues remain even with the updated results, especially about the decreased margin over fp32 results after implementing batch inference for the baseline, and the inconsistency in the observed speedup as reported in previous work "Low-bit Quantization for Deep Graph Neural Networks with Smoothness-aware Message Propagation". Therefore, while I appreciate the authors' efforts putting into the rebuttal, some weakness remains, and I hope to see better comparisons with existing works in the next version

**Additional Comments On Reviewer Discussion:**

The reviewers (except PWvf) made concrete points about the weakness of the paper. While the reviewers provided convincing experimental results for most of the points, some concerns remain regarding the inconsistency of the speedups reported by previous papers (as in the discussions of reviewer fkMV).

---

### Decision · Program_Chairs · 2025-01-22

Reject